# NAUTILUS 🛸: A Large Multimodal Model for Underwater Scene Understanding

**Wei Xu**[1*], **Cheng Wang**[1*], **Dingkang Liang**[1], **Zongchuang Zhao**[1],
**Xingyu Jiang**[1], **Peng Zhang**[2], **Xiang Bai**[1†]

[1]Huazhong University of Science and Technology
[2]National University of Defense Technology
{wxu2023, cwang666, dkliang, xbai}@hust.edu.cn

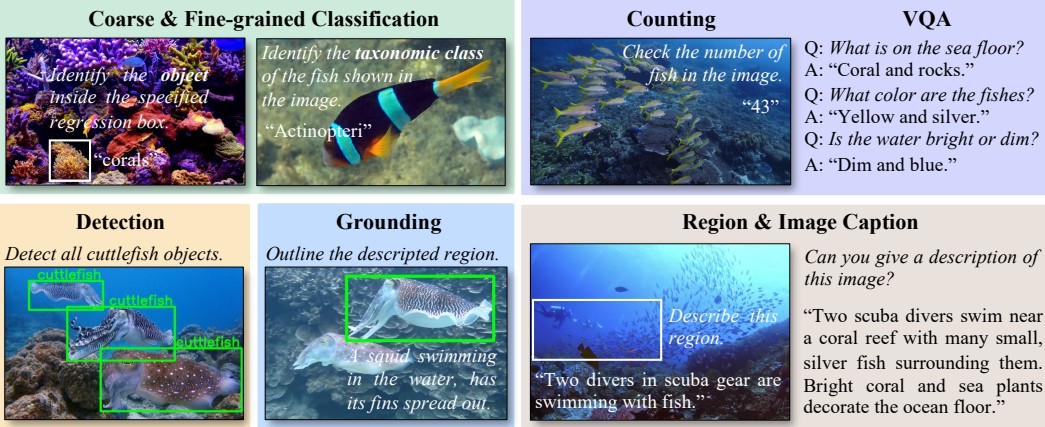

Figure 1: The underwater environment presents a visually rich and dynamically evolving landscape. NAUTILUS addresses eight diverse underwater tasks, encompassing coarse-grained classification, fine-grained classification, counting, visual question answering (VQA), detection, grounding, region caption, and image caption, enabling comprehensive understandings across multiple granularities.

## Abstract

Underwater exploration offers critical insights into our planet and attracts increasing attention for its broader applications in resource exploration, national security, etc. We study the underwater scene understanding methods, which aim to achieve automated underwater exploration. The underwater scene understanding task demands multi-task perceptions from multiple granularities. However, the absence of large-scale underwater multi-task instruction-tuning datasets hinders the progress of this research. To bridge this gap, we construct NautData, a dataset containing $1.45\,$M image-text pairs supporting eight underwater scene understanding tasks. It enables the development and thorough evaluation of the underwater scene understanding models. Underwater image degradation is a widely recognized challenge that interferes with underwater tasks. To improve the robustness of underwater scene understanding, we introduce physical priors derived from underwater imaging models and propose a plug-and-play vision feature enhancement (VFE) module, which explicitly restores clear underwater information. We integrate this module into renowned baselines LLaVA-1.5 and Qwen2.5-VL and build our underwater LMM, NAUTILUS. Experiments conducted on the NautData and public underwater

---

[*]Equal contribution. [†]Corresponding author.

39th Conference on Neural Information Processing Systems (NeurIPS 2025).

datasets demonstrate the effectiveness of the VFE module, consistently improving the performance of both baselines on the majority of supported tasks, thus ensuring the superiority of NAUTILUS in the underwater scene understanding area. Data and models are available at https://github.com/H-EmbodVis/NAUTILUS.

# 1 Introduction

The underwater world plays a pivotal role in global well-being, as it covers more than 70% of the Earth's surface and encompasses the largest ecosystem on our planet [14, 7, 39, 46]. Advancing underwater scene understanding methods facilitates automated underwater robot exploration [48], benefiting sufficient environmental protection [23] and resource development [57]. Comprehensive underwater scene understanding comprises both perception (e.g., detection and counting) and semantic understanding tasks (e.g., region and image caption). However, most underwater methods [66, 49, 59] are typically tailored to a specific task, limiting their understanding of underwater scenes to a task-specific perspective.

Recent achievements in the general domain have driven the advancement of specialized LMMs in understanding tasks [62, 52], leading to prominent applications in fields like autonomous driving [41], embodied intelligence [16], and document understanding [26]. The promising abilities of large multimodal models (LMMs) provide a considerable solution for the underwater scene understanding area. Nonetheless, we empirically find that *directly adopting general LMMs in underwater scenes cannot serve as an ideal solution*, as they face inherent challenges arising from 1) the significant domain shift between in-air and underwater data, and 2) image degradation due to light scattering and absorption in water.

OceanGPT [8] infuses the richness of knowledge into a large language model (LLM), uncovering its potential in the field of ocean science while failing to interpret multimodal inputs. MarineGPT [65] empowers the LLMs to sense vision-language information and has been the first-ever publicly available underwater LMM. As shown in Tab. 1, this pioneering work constructs a large-scale underwater vision-language dataset for instruction tuning while only focusing on the image-level understanding, neglecting the hierarchical underwater scene information. We take a valuable step in constructing NautData, an underwater instruction-following dataset containing 1.45 million question-answer pairs that cover eight diverse underwater tasks. It establishes a solid foundation for the development of underwater LMMs, bypassing the in-air domain shift prevalent in most current instruction-tuning datasets.

The image degradation problem hinders reliable underwater scene understanding. Most methods leave this challenge to the training process, driving the models to learn underwater representations on their own, which could be sub-optimal due to the intricate complexity of underwater conditions. To bridge this gap, we propose a plug-and-play vision feature enhancement (VFE) module that explicitly removes noise responses introduced by image degradation and enhances the understanding performance of underwater LMMs. Specifically, the underwater imaging model provides a physical representation of image degradation in underwater scenarios [67, 61, 3, 38], emphasizing backscattering from the surroundings as a primary interference. We adopt a dark pixel prior to quantifying the intensity of backscattering, paving the way for removing its adverse effects. The optical medium influences imaging quality, with underwater imaging facing significant light absorption compared to in-air imaging, and is another crucial factor contributing to underwater image degradation. To address this, we extract depth information to restore the scene signals attenuated by medium absorption. The VFE module can be flexibly employed in general LMMs, and we integrate it into two renowned baselines, LLaVA-1.5 [32] and Qwen2.5-VL [6], to build our underwater LMM NAUTILUS.

We evaluate the NAUTILUS and prestigious LMMs [32, 56, 10, 6] on the NautData to analyze their underwater scene understanding performance in a supervised manner. Then, we directly evaluate them on MarineInst [63], a recent large-scale vision-language underwater dataset. This zero-shot experiment indicates the generalization capabilities of our method.

This work contributes to three aspects: **1)** We construct NautData, a large-scale underwater instruction-following dataset containing 1.45 M image-text pairs, enabling developments and evaluations of underwater LMMs. **2)** We build the first eight-task underwater LMM NAUTILUS, achieving underwater scene understanding from image, region, and object levels. It empowers comprehensive underwater scene understanding through aggregating hierarchical scene information. **3)** We design a

Table 1: The comparisons between NautData and recent underwater vision-language datasets. Our dataset is more comprehensive for eight-task annotations involving understandings at three granularities, making it a valuable contribution to the community.

| Datasets | Reference | Supported tasks | | | | | | | | Granularity | | | QA pairs | Open source |
|---|---|---|---|---|---|---|---|---|---|---|---|---|---|---|
| | | VQA | Detection | Classification | | Grounding | Caption | | Counting | Img. | Reg. | Obj. | | |
| | | | | Coarse | Fine | | Image | Region | | | | | | |
| MarineGPT [65] | arXiv 23 | - | - | - | - | - | ✓ | - | - | ✓ | - | - | 1.12 M | Not Avail. |
| MarineInst20M [63] | ECCV 24 | - | ✓ | - | - | ✓ | ✓ | - | - | ✓ | - | ✓ | 20 M | Part. Avail.[a] |
| CoralMask [64] | CVPR 24 | - | ✓ | - | - | ✓ | - | - | - | - | - | ✓ | 46.61 K | Avail. |
| AquaticCLIP [5] | arXiv 25 | - | - | - | - | - | ✓ | - | - | ✓ | - | - | 2 M | Not Avail. |
| SynTIDE [31] | CVPR 25 | - | - | - | - | - | ✓ | - | - | ✓ | - | - | 54.51 K | Avail. |
| NautData (**ours**) | - | - | ✓ | ✓ | ✓ | ✓ | ✓ | ✓ | ✓ | ✓ | ✓ | ✓ | 1.45 M | Avail. |

[a] Current public version contains 2.2 M QA pairs.

plug-and-play VFE module motivated by a physical underwater imaging model. It restores degraded information explicitly in the feature space. Experiments on renowned baselines demonstrate its effectiveness on all the annotated tasks.

## 2 Related Work

**Underwater Vision-Language Analysis.** Pioneering studies [34, 68, 32] have aligned linguistic and visual representations using projection layers, enabling LLMs to process multimodal information and advance into LMMs. Through visual instruction tuning, LMMs have presented impressive performance in both general-purpose and domain-specific areas. For instance, LLaVA-NeXT [33], InternVL [11], and MiniGPT-v2 [9] have achieved remarkable success in the general domain by advancing LMMs through improvements in data, vision encoder structures, and training strategies. PaLM-E [16] and Dolphins [36] find LMMs effective in embodied intelligence and autonomous driving, respectively. MarineGPT [65] trains LMMs with marine science knowledge, focusing primarily on image-level understanding while lacking attention to region- and object-level [37, 30] scene information. AquaticCLIP [5] employs contrastive learning-based pretraining on aquatic image-text pairs to align the underwater image and text representations. The AquaticCLIP enhances performance on downstream underwater tasks while lacking the capability to follow instructions directly. MarineInst [63] localizes objects and generates linguistic descriptions for each of them. The object-level vision and text responses contribute to detailed marine image analysis. CoralSCOP [64] can be driven by both vision and text prompts, enabling the mask generation of corals described by users. Despite increasing attention, research on underwater LMMs remains limited and requires further efforts to achieve underwater vision-language dialogues. This paper advances this field by exploring the capabilities of LLMs to deliver hierarchical underwater scene understandings.

**Underwater Image Enhancement.** Underwater images often exhibit poor visibility, low contrast, and severe color distortions, primarily due to light absorption and scattering in aquatic environments. To address these issues, underwater image enhancement methods aim to mitigate such degradations and restore visual quality comparable to in-air images. Conventional methods employ typical image augmentation strategies, such as histogram stretching [15, 60] and image fusion [20] to enhance the underwater images. These methods are easy to deploy, but they suffer from the generalization limitations of handcrafted features. Another direction of underwater image enhancement research [19, 54, 17] explores deep learning-based methods, particularly those leveraging generative adversarial networks (GANs) to improve the quality of underwater images. However, these methods are hindered by the inadequate availability of high-quality training data, as collecting underwater images is often constrained by the complexity of underwater environments, high equipment costs, and the challenges associated with accurate data annotation. Physical model-based methods [67, 61, 51] offer a feasible solution to these problems. They reduce the search space for parameters by incorporating handcrafted priors, thus mitigating dependency on large-scale training data. However, directly applying image enhancement to underwater images may result in information loss, thereby limiting the effectiveness in underwater scene understanding, as demonstrated in our experiments. Building upon the previous discussion, we innovatively introduce an enhancement in feature space within an LMM, providing an efficient solution to extract underwater visual information.

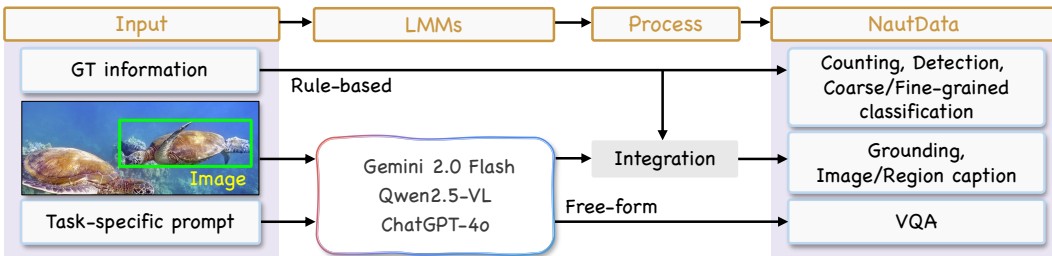

Figure 2: Illustration of the data construction framework. Eight tasks are involved, and the data generation process is tailored to each task. Rule-based generation utilizes predefined templates to generate question-answer pairs. Integration generation integrates question-answer pairs using both templates and outputs from LMMs. Free-form generation enables LMMs to construct questions and answers based on the content they focus on.

## 3 Dataset Construction

Underwater vision-language datasets [65, 63] lack multi-granular and multi-task annotations. To address this gap, we construct NautData, which provides a pioneering resource for advancing research on underwater scene understanding. The distinct advantages of NautData lie in three aspects: (1) extensive underwater information encompassing image-, region-, and object-level understanding; (2) diverse conversational structures, including both rule-based and free-form content; and (3) significant scale, comprising 158 K images and 1.45 M QA pairs.

**Attributes of Individual Entities.** Public underwater object detection datasets such as RUOD [18], Deepfish [47], and Brackish [42] provide bounding-box annotations reflecting the position information of entities. As shown in Fig. 2, we employ a rule-based procedure to reformulate the labeled coordinates into a linguistic answer "$\texttt{Class:}[x_1, y_1, x_2, y_2]\texttt{,...,}\texttt{ Class:}[x_1, y_1, x_2, y_2]$". Then, we concatenate it with a pre-defined question "$\texttt{Detect all underwater objects in the image.}$" to create an object-level perception conversation.

Classifying aquatic targets constitutes an expert-level task due to the specialized nature of underwater knowledge. We design both coarse-grained and fine-grained classification conversations, facilitating knowledge sharing in this domain. Specifically, a standard detection task distinguishes underwater objects in coarse-grained categories, e.g., fish, turtles, or reefs. FishNet [25] further fine-grains the fish into 8 taxonomic classes. We pre-define questions with corresponding detection categories as answers to deliver coarse-grained classification conversations. Subsequently, for the FishNet dataset, we couple questions with ground-truth taxonomic class names as answers to construct fine-grained classification dialogs.

Apart from the formalized annotations of positions and categories, we employ LMMs [50, 6, 40] to generate textual descriptions, which could contain rich individual properties, such as color, texture, and shape. For instance, we inject object coordinates in textual form into prompts, directing concentrations of LMMs on target local regions. The generated descriptions are subsequently integrated with these coordinates to construct grounding conversations.

**Annotations on Regional Groups.** Underwater species often exhibit collective behaviors essential for understanding their survival and ecosystem dynamics. IOCfish5K [49] is a densely distributed underwater object counting dataset, averaging 117 targets per image. We design conversations on group counts and behaviors to enhance regional understanding. Specifically, we first couple pre-defined questions with ground-truth counts as answers, prompting models to regress numerical results directly. We then convert this regression task into a single-choice question by randomly selecting intervals from $\{5, 50, 100\}$ to construct four-term arithmetic sequences that include the ground-truth count. Additionally, we employ LMMs [50, 6, 40] to generate descriptions for regional groups, which may describe collective behaviors and relationships. These descriptions are integrated with pre-defined questions to create region caption dialogs.

**Descriptions for Holistic Semantics.** Image-caption pairs have been demonstrated to significantly benefit LMMs in aligning vision and language modalities [6, 34, 24]. We feed collected images into LMMs [50, 6, 40] to obtain a holistic understanding of the given scenes. The outputs are coupled

with pre-defined questions to construct image caption conversations. Underwater images record rich information rarely discussed in existing datasets, such as brightness and geomorphology. We employ LMMs [50, 6, 40] to generate free-form visual question-answering (VQA) conversations discussing these elements.

During the generation process utilizing LMMs, we first employ Gemini 2.0 Flash to produce initial outputs. Subsequently, these outputs are evaluated by Qwen2.5-VL-72B, and any responses identified as low-quality are replaced with newly generated answers. Leveraging the constructed NautData, we collect image-text pairs to develop the NautData test set, which is further assessed by GPT-4o. Answers flagged as low-quality undergo additional manual verification by our research team.

The current NautData test set comprises $3,920$ images paired with $7,916$ question-answering (QA) examples. These QA pairs encompass a diverse range of tasks, including image and region captioning, coarse-grained and fine-grained classification, grounding, detection, counting, and visual question answering (VQA). This comprehensive benchmark is designed to facilitate a rigorous evaluation of methods in underwater scene understanding, thereby promoting further advancements in this field.

## 4 Methodology

Underwater environments, characterized by intricate conditions and entities of diverse colors, shapes, and scales, require multi-grained perception to achieve comprehensive scene understanding. NAUTILUS is the first model empowering vision-language conversations spanning image-, region-, and object-level underwater scene understanding tasks, potentially facilitating seamless human-computer interaction and underwater knowledge sharing.

To illustrate the design of NAUTILUS, we first review the physical underwater imaging model to depict the motivation of explicitly dealing with underwater image degradation. Then, we detail the implementation of the model architecture involving the overall framework and the vision feature enhancement (VFE) module.

### 4.1 Preliminaries

Underwater imaging model [67, 61, 3, 38] typically formulates the captured underwater image $I_c$ as the combination of the direct reflection $D_c$ from the underwater subjects and the backscattering $B_c$ from the surroundings:

$$I_c = D_c + B_c, \quad D_c = J_c e^{-\beta_c(z) \cdot z}, \tag{1}$$

where $J_c$ is the original color at the $c$-th channel without light absorption during propagation. The underwater imaging model assumes one attenuation coefficient $e^{-\beta_c(z) \cdot z}$ for each color, decreasing exponentially with the imaging distance $z$. The $\beta_c(z)$ is an unavailable parameter related to data collection conditions, also primarily regulated by imaging depth. Inspired by this physical imaging model, we attempt to restore the representations of $J_c$ as enhanced vision features:

$$J_c = \frac{(I_c - B_c)}{e^{-\beta_c(z) \cdot z}}. \tag{2}$$

In practice, we fit the attenuation coefficient by employing a learning module with depth information as inputs. Furthermore, we introduce a dark pixel prior [67] to quantify the impact of backscattering $B_c$. For instance, due to the surrounding backscattering in underwater environments, dark pixels often lose their original black appearance and instead exhibit a blue-green color. The dark pixel prior emphasizes that these distorted pixel values located at dark pixels reflect influences of backscatterings. Drawing on this analysis, we localize the dark pixels of a given image and regard their responses as backscattering intensities. Subsequently, simply substitute the obtained parameters in Eq. 2 to complete the feature enhancement.

The underwater imaging model offers a physical principle to regularize the learning phase, which we consider an explainable and sufficient optimization direction for model design.

### 4.2 Model Architecture

As shown in Fig. 3, the framework of NAUTILUS primarily consists of an image encoder $\mathcal{I}_v$, a depth encoder $\mathcal{I}_d$, a vision-to-language projector $\mathcal{P}_{v-l}$, a VFE module, and an LLM. Given an underwater

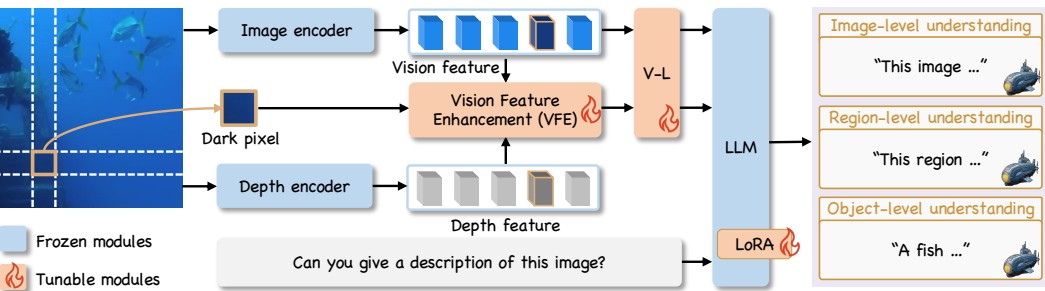

Figure 3: The framework of NAUTILUS. Inspired by underwater physical priors [67, 3, 38], we sample dark pixels to quantify the responses of underwater degradation. The vision feature enhancement (VFE) module improves underwater LMMs with depth information as auxiliary information. Outputs of the image encoder and the VFE module are fed into an LLM to facilitate multimodal processing.

image $x$, the image encoder extracts vision features $v = \mathcal{I}_v(x)$. While employing LLaVA-1.5 [32] as the baseline, we employ a CLIP ViT-L/14 [45] with a base resolution of 336 as the image encoder. The vision-to-language projector is a multi-layer perceptron aligning vision and language representations. It empowers the LLM Vicuna-v1.5 [12] to reason about visual and textual information.

Motivated by the underwater imaging models [67, 3, 38], a physical prior illustrates that the imaging distance closely influences the degree of image degradation. In particular, subjects farther away from the camera suffer more pronounced color degradation, deviating further from their original appearances. Therefore, we adopt a frozen Depth Anything V2 [55] encoder to extract depth features $d = \mathcal{I}_d(x)$ from given scenes. In addition, inspired by the dark pixel prior introduced in Sec. 4.1, it is reasonable to measure the backscattering influence by analyzing the responses of dark pixels. In practice, as pixels are processed in a patch-wise manner, we identify the $k$-th image patch that exhibits the lowest average RGB value and treat the pixels within this patch as dark pixels. Subsequently, we feed the vision feature, index $k$, and depth feature into the VFE module $\mathcal{M}$ to obtain an enhanced vision feature $v_e = \mathcal{M}(v, k, d)$. We would like to emphasize that both the original and enhanced vision features are essential for understanding underwater scenes. Specifically, on the one hand, the degradations in the original vision feature reflect authentic underwater environments, facilitating the study of real ecosystems. On the other hand, the restored information in the enhanced vision feature reduces the adverse effects of imaging conditions, enabling reliable underwater perceptions. Therefore, we feed them forward in parallel. And due to the homologous representations of the two features, we utilize a shared projector to align them with the linguistic modality. This process can be formulated as follows:

$$\hat{v} = \mathcal{P}_{v-l}(v), \quad \hat{v_e} = \mathcal{P}_{v-l}(v_e), \tag{3}$$

where $\hat{v}$ and $\hat{v_e}$ are aligned vision features derived from $v$ and $v_e$, respectively. Afterward, we employ the LLM to integrate user instructions and vision information, finally achieving multi-granular underwater scene understandings from multi-task aspects.

### 4.3 Vision Feature Enhancement

According to Eq. 1, the underwater vision feature enhancement comprises two steps: 1) removing backscattering and 2) restoring light absorptions. We reveal the entire process in Fig. 4 and clarify the reasons for taking them.

**Remove Backscattering.** Assuming $v = \{f_{v,i}\}_{i=1}^n \in \mathbb{R}^{n \times d}$, where $n$ and $d$ denote the length and dimension of the vision feature, we regard the $k$-th slice $f_{v,k} \in \mathbb{R}^{1 \times d}$ as a dark token, representing the response of the dark pixels. Simply subtracting it in feature space would filter out responses from the backscattering. However, the dark token is encoded through multiple attention layers, which infuses global semantics beyond the backscattering. Therefore, we further isolate the global semantic responses from this token to estimate a pure backscattering intensity. Specifically, we employ a cross-attention layer with the vision feature as the key and value to aggregate global information into a learnable query. This query is also embedded with a global average feature, which is obtained by applying average pooling over the vision feature, guiding the learnable query to be more familiar with the global semantics. Assuming the output of this cross-attention layer as $q \in \mathbb{R}^{1 \times d}$, the

backscattering responses $s \in \mathbb{R}^{1 \times d}$ can be refined into $f_{v,k} - q$. As the backscattering is added to the whole underwater scene, disturbing every image pixel, we remove backscattering by pixel-wise subtracting it from the vision feature. This process can be simply formulated as $v - s$, where $s$ is broadcast to $\mathbb{R}^{n \times d}$, keeping the same shape as $v$.

**Restore Light Absorption.** According to the underwater imaging model, underwater light is absorbed over imaging distances. We predict an absorption weight $W \in \mathbb{R}^{n \times d}$ using a lightweight multilayer perceptron MLP with the depth feature as an input, i.e., $W = \text{MLP}(d)$. Finally, we obtain the enhanced vision feature $v_e$ by calculating:

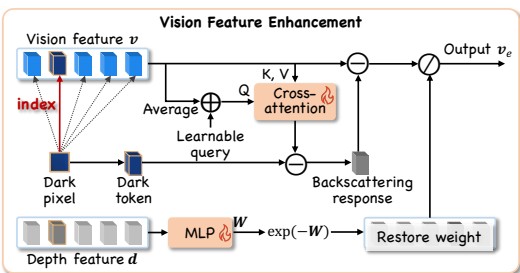

$$v_e = (v - s) \oslash \exp(-W), \qquad (4)$$

where $\oslash$ and $\exp(\cdot)$ denote element-wise division and exponentiation, respectively. Eq. 4 conforms to Eq. 2, explicitly injecting human priors into network structure and regularizing restorations of visual responses in feature space.

Figure 4: The structure of the vision feature enhancement (VFE) module. The inputs consist of the vision feature, the index of the dark pixel, and the depth feature. It outputs enhanced vision features capturing restored underwater information.

## 5 Experiments

We explain the implementation details and conduct experiments to evaluate the performance of our model. As this is a pioneering work towards underwater understanding and perception employing LMMs, we select distinguished general-purpose baselines as counterparts to present comprehensive comparisons.

**Implementation Details.** We enhance vision features to address the native underwater image degradation, as described in Sec. 4.3. To demonstrate how this design benefits current LMMs, we adapt it to LLaVA-1.5 [32] and Qwen2.5-VL [6], two prevalent LMM frameworks. For both of them, we adopt a parameter-efficient fine-tuning (PEFT) strategy [53, 29, 22], and the trainable components are the vision-to-language projector, LoRA [22], and the vision feature enhancement module. In our instruction tuning, we preserve the default hyperparameters of LLaVA-1.5 to pursue optimal performance and ensure a fair comparison with its original implementations. As for Qwen2.5-VL, since the official repository only supports full fine-tuning, we reproduce LoRA fine-tuning, setting the learning rate as $2 \times 10^{-5}$ with the batch size as 16. The LoRA ranks in LLaVA-1.5 and Qwen2.5-VL are set as 128. Unless otherwise specified, we use the 7B variants, except for InternVL-2.5, for which we employ the InternVL-2.5-8B model. Our experiments are conducted on four NVIDIA A800-80GB GPUs, training each model for one epoch, taking around 3 days.

**Datasets.** NautData is the first underwater instruction-following dataset providing eight-task annotations. Experiments are primarily conducted on the NautData. MarineInst20M [63] is a recently impressive underwater vision-language dataset containing high-quality image-caption pairs. Among this dataset, we conduct zero-shot evaluations on its human-annotated part involving the Flickr, Shutterstock, and Gettyimages subsets. IOCfish5k [49] is the unique underwater object-counting dataset among the collected datasets. We evaluate the counting performance on its test set.

### 5.1 Comparison to SOTA Methods

**Compared to Renowned LMMs.** We conduct zero-shot experiments using GPT-4o [40], Qwen2.5-VL-72B [6], and Gemini 2.0 Flash [44] on the NautData test set. As shown in Tab. 2, despite their advanced image understanding capabilities, these methods struggle to achieve satisfactory performance due to the inherent complexity of underwater environments. Fine-tuned open-source baselines achieve significant improvements over the commercial LMMs on all tasks, demonstrating the effectiveness of domain-specific adaptation in underwater scene understanding. The proposed NAUTILUS explicitly addresses underwater image degradation, empowering reliable underwater perceptions under various adverse conditions. Specifically, while employing LLaVA-1.5 [32] as the baseline, our method improves its performance on seven of eight tasks, presenting strong practical potential. Furthermore, while employing Qwen2.5-VL [6] as the base model, the NAUTILUS achieves

Table 2: The comparison of our NAUTILUS and renowned LMMs on the NautData test set. The best results are highlighted in **bold**, and the second-best results are underscored.

| Methods | Reference | Classification | | Caption | | Grounding | | Detection | | VQA |
|---|---|---|---|---|---|---|---|---|---|---|
| | | Coarse | Fine | Image | Region | | | | | |
| | | acc ↑ | acc ↑ | METEOR ↑ | METEOR ↑ | mIoU ↑ | PR@0.5 ↑ | mAP ↑ | mAP@0.5 ↑ | METEOR ↑ |
| *Zero-shot experiments* | | | | | | | | | | |
| GPT-4o [40] | - | 55.2 | 54.4 | 0.179 | 0.148 | 14.2 | 4.3 | 0.3 | 1.4 | 0.242 |
| Qwen2.5-VL-72B [6] | - | 55.2 | 54.2 | 0.171 | 0.126 | 42.3 | 46.4 | 8.8 | 14.7 | 0.222 |
| Gemini 2.0 Flash [44] | - | 55.5 | 54.3 | 0.185 | 0.141 | 21.2 | 20.6 | 2.4 | 7.8 | 0.223 |
| *Instruction-tuning experiments* | | | | | | | | | | |
| MiniGPTv2 [9] | arXiv 23 | 80.0 | 90.0 | 0.204 | 0.178 | 47.0 | 51.0 | 6.9 | 12.9 | 0.372 |
| mPLUG-Owl3 [56] | arXiv 24 | **91.9** | 92.0 | 0.219 | **0.207** | 41.1 | 45.7 | 10.3 | 23.3 | **0.383** |
| InternVL-2.5 [10] | arXiv 24 | 91.3 | 90.4 | 0.208 | 0.195 | 50.4 | 54.6 | 18.3 | 30.5 | 0.382 |
| LLaVA-1.5 [32] | CVPR 24 | 90.0 | 89.8 | 0.208 | 0.189 | 43.5 | 48.2 | 9.8 | 19.0 | 0.359 |
| Qwen2.5-VL [6] | arXiv 25 | 85.3 | 88.2 | 0.222 | 0.196 | 52.5 | 57.6 | 24.5 | 41.7 | 0.380 |
| NAUTILUS(LLaVA-1.5) | - | 91.0(+1.0) | 89.9(+0.1) | 0.208(+0.000) | 0.191(+0.002) | 46.2(+2.7) | 52.2(+4.0) | 11.1(+1.3) | 20.9(+1.9) | 0.365(+0.006) |
| NAUTILUS(Qwen2.5-VL) | - | 90.3(+5.0) | **93.8**(+5.6) | **0.223**(+0.001) | 0.199(+0.003) | **53.8**(+1.3) | **58.8**(+1.2) | **25.8**(+1.3) | **45.3**(+3.6) | 0.381(+0.001) |

the best performance on four tasks, including fine-grained classification, image caption, grounding, and detection, demonstrating remarkable underwater scene understanding capabilities.

**Group Perception.** The object counting task provides insight into group behaviors. We evaluate NAUTILUS on the underwater object counting task to assess its capability for group understanding. As shown in Tab. 3, we follow the official data splits of IOCfish5k [49] to construct training and test subsets. Our methods outperform other LMMs by at least 0.7 MAE and 0.3 RMSE, delivering 8.0 MAE and 15.9 RMSE improvements on LLaVA-1.5, which exhibits impressive group perception performance. Nevertheless, there is a slight performance drop in the accuracy metric, which we attribute to the challenges of multi-task optimization. Specifically, the single-choice question is a text classification task substantially different from the count regression task in both objectives and output formats, introducing further challenges.

**Generalization.** After fine-tuning the well-established baselines on the NautData, we directly evaluate their grounding performance on the MarineInst20M [63]. The grounding task necessitates instance-level understanding and text comprehension capabilities, which reflect the ability to achieve fine-grained understanding of underwater scenes. As shown in Tab. 4, our method improves the LLaVA-1.5 [32] and Qwen2.5-VL [6] by 0.6 and 0.4 PR@0.5, respectively, demonstrating its generalization ability across domains and models.

Table 3: Counting accuracy on the IOCfish5k [49] test set.

| Methods | Reference | MAE ↓ | RMSE ↓ | acc ↑ |
|---|---|---|---|---|
| *Zero-shot experiments* | | | | |
| GPT-4o [40] | - | 51.2 | 94.0 | 55.4 |
| Gemini 2.0 Flash [44] | - | 50.2 | 123.3 | 50.7 |
| Qwen2.5-VL-72B [6] | - | 49.8 | 124.7 | 58.8 |
| *Instruction-tuning experiments* | | | | |
| MiniGPTv2 [9] | arXiv 23 | 55.0 | 139.6 | 43.0 |
| mPLUG-Owl3 [56] | arXiv 24 | 36.2 | 102.9 | 70.5 |
| InternVL-2.5 [10] | arXiv 24 | 39.1 | 95.0 | 69.2 |
| LLaVA-1.5 [32] | CVPR 24 | 59.2 | 151.9 | 61.7 |
| Qwen2.5-VL [6] | arXiv 25 | 31.6 | 96.7 | **70.8** |
| NAUTILUS(LLaVA-1.5) | - | 51.2(+8.0) | 136.0(+15.9) | 62.6(+0.9) |
| NAUTILUS(Qwen2.5-VL) | - | **30.9**(+0.7) | **93.7**(+3.0) | 70.7(-0.1) |

Table 4: Zero-shot grounding on the Marine-Inst20M [63].

| Methods | Reference | mIoU ↑ | PR@0.5 ↑ |
|---|---|---|---|
| MiniGPTv2 [9] | arXiv 23 | 36.6 | 38.3 |
| mPLUG-Owl3 [56] | arXiv 24 | 32.9 | 36.7 |
| InternVL-2.5 [10] | arXiv 24 | 37.6 | 38.4 |
| LLaVA-1.5 [32] | CVPR 24 | 29.6 | 33.6 |
| Qwen2.5-VL [6] | arXiv 25 | 38.0 | 39.2 |
| NAUTILUS(LLaVA-1.5) | - | 30.7(+1.1) | 34.2(+0.6) |
| NAUTILUS(Qwen2.5-VL) | - | **38.3**(+0.3) | **39.6**(+0.4) |

## 5.2 Analysis and Ablation

Fig. 5 presents qualitative visualizations of NAUTILUS across eight underwater tasks. Our model responds to user instructions and outputs multi-granularity results for underwater scene understanding, demonstrating its versatility and effectiveness. In this section, we perform ablation studies and provide an analysis of the key insights. Models in Tab. 5 and Tab. 6 are trained on one-third of the

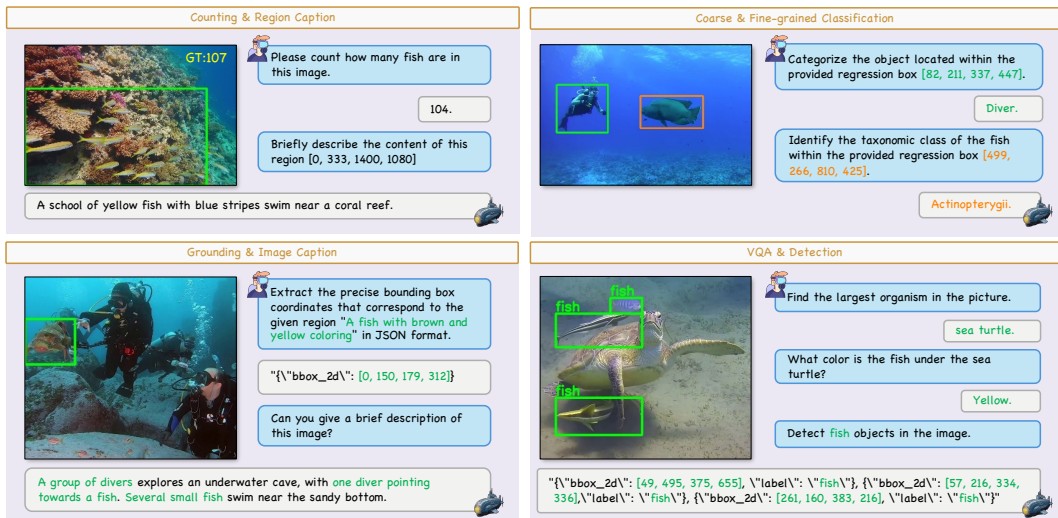

Figure 5: Qualitative results on underwater scene understanding. NAUTILUS perceives image-, region-, and object-level information while addressing eight tasks. Our underwater LMM exhibits remarkable multimodal instruction-following performance, serving as a meaningful contribution to this field.

Table 5: Ablation on components. We employ Qwen2.5-VL [6] as the baseline model and sequentially add our modules. Evaluations are conducted on eight tasks to provide a comprehensive analysis.

| Baseline | Depth encoder | Vision Feature Enhancement | | Classification | | Caption | | Counting | Grounding | Detection | VQA |
| | | Restore light absorption | Remove backscattering | Coarse | Fine | Image | Region | | | | |
| | | | | acc ↑ | acc ↑ | METEOR ↑ | METEOR ↑ | MAE ↓ | PR@0.5 ↑ | AP@0.5 ↑ | METEOR ↑ |
| ✔ | - | - | - | 87.9 | 89.1 | **0.222** | 0.197 | 36.8 | 55.4 | 35.9 | 0.367 |
| ✔ | ✔ | - | - | 89.5 | 89.1 | 0.218 | 0.194 | 37.2 | 55.0 | **36.4** | 0.369 |
| ✔ | ✔ | ✔ | - | 85.7 | 91.2 | 0.220 | 0.195 | **36.2** | 53.9 | 34.2 | 0.372 |
| ✔ | ✔ | ✔ | ✔ | **90.0** | **91.4** | **0.222** | **0.198** | 36.4 | **55.9** | 36.2 | **0.373** |

NautData training set and evaluated on the full NautData test set. Tab. 7 presents the performance of models fine-tuned on the complete NautData training set when evaluated under various degraded conditions.

**Design of the VFE Module.** As shown in Tab. 5, we evaluate the effectiveness of our components. The depth encoder provides rich depth information, which is expected to enhance the LMM's understanding of underwater scenes. However, due to the discrepancy in feature distribution, simply adding a depth encoder results in performance degradation on four tasks. In contrast, we utilize depth information to enhance light absorption and achieve improvements in five tasks compared to the second line, emphasizing the feasibility of feature fusion guided by the underwater imaging model. Subsequently, we remove backscattering to complete the restoration phase, which further surpasses the third line on five tasks, highlighting the benefits of each component.

**Enhancement in Feature Space.** We train the Qwen2.5-VL baseline with all images enhanced by Reti-Diff [21], SMDR-IS [58], and CCL-Net [35], three state-of-the-art underwater image restoration methods, to assess the benefits of image enhancement. As shown in Tab. 6, there are consistent performance drops in the coarse-grained classification, image caption, region caption, and detection tasks. We attribute these inferior results to the information loss introduced by image enhancement during image pre-processing. In contrast, feature enhancement preserves the image's original information to the greatest extent, leading to higher reliability and thus deserves wide application.

**Evaluation under Degraded Conditions.** Underwater environments often exhibit distinct characteristics in lighting and turbidity. To evaluate the robustness of NAUTILUS under such domain shifts, we assess its performance across varying environmental conditions. Specifically, we employ Gemini 2.5 Flash [13] to categorize the NautData test set based on lighting conditions (low-light,

Table 6: Ablation on components. We employ Qwen2.5-VL [6] as the baseline model and sequentially add our modules. Evaluations are conducted on eight tasks to provide a comprehensive analysis.

| Methods | Classification | | Caption | | Counting | Grounding | Detection | VQA |
|---|---|---|---|---|---|---|---|---|
| | Coarse | Fine | Image | Region | | | | |
| | acc ↑ | acc ↑ | METEOR ↑ | METEOR ↑ | MAE ↓ | PR@0.5 ↑ | AP@0.5 ↑ | METEOR ↑ |
| Baseline | 87.9 | 89.1 | **0.222** | 0.197 | 36.8 | 55.4 | 35.9 | 0.367 |
| +Reti-Diff [21] | 87.3 | 91.1 | 0.221 | 0.194 | **36.3** | 55.5 | 35.0 | 0.370 |
| +SMDR-IS [58] | 86.8 | 86.3 | 0.220 | 0.195 | 36.5 | 54.4 | 31.4 | 0.371 |
| +CCL-Net [35] | 82.5 | 87.3 | 0.220 | 0.193 | 37.8 | 54.2 | 32.9 | 0.365 |
| +VFE (**ours**) | **90.0** | **91.4** | **0.222** | **0.198** | 36.4 | **55.9** | **36.2** | **0.373** |

Table 7: Ablation on degraded conditions. We divide the NautData test set into subsets based on the degradations of data and evaluate grounding performance on these subsets using the PR@0.5 metric.

| Methods | Reference | Low-light | Normal-light | Green-tinted | Blue-tinted | Turbid | Clear |
|---|---|---|---|---|---|---|---|
| MiniGPTv2 [9] | arXiv 23 | 44.7 | 52.8 | 47.0 | 52.7 | 46.6 | 53.5 |
| mPLUG-Owl3 [56] | arXiv 24 | 41.0 | 46.8 | 42.7 | 46.7 | 41.2 | 48.0 |
| InternVL-2.5 [10] | arXiv 24 | 52.7 | 55.1 | 53.4 | 55.0 | 52.7 | 55.6 |
| LLaVA-1.5 [32] | CVPR 24 | 44.1 | 49.1 | 43.5 | 50.1 | 41.9 | 51.8 |
| NAUTILUS(LLaVA-1.5) | - | 51.6(+7.5) | 52.0(+2.9) | 51.8(+8.3) | 51.9(+1.8) | 50.0(+8.1) | 53.1(+1.3) |
| Qwen2.5-VL [6] | arXiv 25 | 56.9 | 58.5 | 56.5 | 58.9 | 53.4 | 59.3 |
| NAUTILUS(Qwen2.5-VL) | - | **58.5**(+1.6) | **58.7**(+0.2) | **57.7**(+1.2) | **59.2**(+0.3) | **55.4**(+2.0) | **60.8**(+1.5) |

normal-light), water turbidity (turbid, clear), and color casts (green-tinted, blue-tinted). As shown in Tab. 7, NAUTILUS demonstrates exceptional robustness, particularly under challenging conditions. Compared to the baseline LLaVA-1.5, NAUTILUS achieves substantial improvements of 7.5, 8.3, and 8.1 PR@0.5 in low-light, green-tinted, and turbid scenarios, respectively. Even under less challenging conditions, NAUTILUS maintains consistent performance gains. Compared to the baseline Qwen2.5-VL, equipped with strong grounding performance, NAUTILUS still achieves notable improvements for at least 1.2 PR@0.5 facing degradations. These results demonstrate the remarkable robustness and practical applicability of NAUTILUS across diverse underwater environments.

## 6    Conclusion

We introduce physical principles to regularize the model architecture and propose a vision feature enhancement (VFE) module. Integrating this module into renowned LLaVA-1.5 and Qwen2.5-VL, we develop NAUTILUS, the first underwater LMM addressing underwater image degradation explicitly. Furthermore, we construct NautData to bridge this gap of the absent underwater multi-task instruction-following dataset. Experiments on both NautData and public underwater benchmarks demonstrate the effectiveness of the VFE module, consistently improving baselines across almost all tasks. Comparisons with current state-of-the-art methods further highlight the superiority of NAUTILUS, establishing our work as a valuable contribution to the community.

**Limitation.** The vast diversity of underwater environments and species poses substantial challenges for exhaustively representing all relevant categories and scenarios in current datasets. Therefore, underwater scene understanding algorithms must possess open-vocabulary or few-shot learning capabilities to generalize effectively to novel and unseen cases, which is under-explored in our work.

## Acknowledgements

This work was supported by the NSFC (U2341227 and 62225603).

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

# Appendix

In this appendix, we provide additional content to complement the main manuscript:

- Appendix A: Detailed information about the dataset, including 1) Distributions of the NautData. 2) Specific prompts for each task.
- Appendix B: More results, including 1) Comparisons employing more metrics on specific tasks. 2) More ablation studies. 3) More visualizations.
- Appendix C: Add discussions about our core insights.

## A    Detailed Information of the NautData

### A.1    Dataset Characteristics

Table I: Distribution of QA pairs generated from different datasets. "Images" indicates the number of images collected from each dataset. "Proportion" refers to the percentage of QA pairs derived from each dataset relative to the total number of QA pairs.

| Dataset | Images | QA pairs | Proportion |
|---|---|---|---|
| USIS10k [28] | 10,632 | 179,772 | 12.5% |
| UIIS [27] | 4,628 | 123,492 | 8.6% |
| RUOD [18] | 14,000 | 326,068 | 22.6% |
| Deepfish [47] | 4,505 | 97,007 | 6.7% |
| Brackish [43] | 12,444 | 245,358 | 17.0% |
| IOCfish5k [49] | 5,637 | 64,037 | 4.4% |
| UVOT-400 [4] | 9,064 | 169,988 | 11.8% |
| Aquarium [1] | 638 | 15,076 | 1.0% |
| Underwater Trash [2] | 5,130 | 33,429 | 2.3% |
| FishNet [25] | 94,806 | 188,450 | 13.1% |

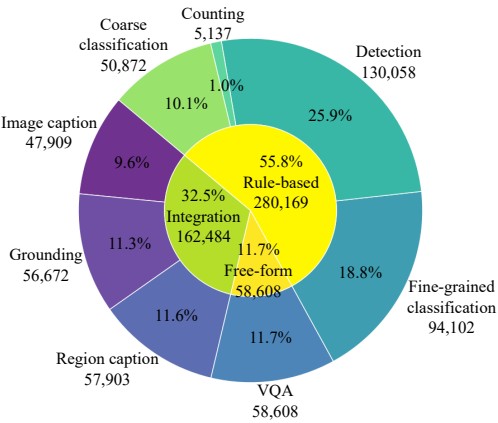

Figure I: The distribution of images involved in different tasks.

We collect source data from 10 public underwater datasets and present the number of QA pairs generated from each dataset in Tab. I. The RUOD [18] and Aquarium [1] contribute to the most and least QA pairs, with the numbers of $326,068$ and $15,076$, respectively. The NautData provides rich annotations covering 8 perception and understanding tasks. As shown in Fig. I, we count images involved in different tasks. The counting task merely contains images from the IOCfish5k [49] dataset, presenting a $1.0\%$ proportion of the total images. Most images are used to generate rule-based multi-modal instructions, constituting $55.8\%$ of all images.

### A.2    Generation Prompts

We display the task-specific prompts in Tab. VII. These prompts guide LMMs in generating descriptions at a specified granularity and in a prescribed format. The generated descriptions are subsequently combined with pre-defined questions to construct QA pairs. In particular, we provide pre-defined question templates in Tab. VIII.

## B    More Results

### B.1    Detailed Results on Each Task

We perform comparisons on all supported tasks to provide a comprehensive evaluation of the proposed NAUTILUS. As shown in Tab. II and Tab. IV, our model ranks first on five of six metrics, highlighting its superior object-level perception capability. Tab. III illustrates the underwater image and region caption performance of current SOTAs. Our method presents superior performance on the image caption task, surpassing other methods on both the CIDEr and METEOR metrics. However, it

Table II: Comparisons on the detection task.

| Method | mAP@0.75 ↑ | AR@100 ↑ | mAP ↑ |
|---|---|---|---|
| *Zero-shot experiments* | | | |
| GPT-4o [40] | 0.03 | 0.99 | 0.30 |
| Qwen2.5-VL-72B [6] | 8.58 | 14.04 | 8.79 |
| Gemini 2.0 Flash [44] | 0.64 | 5.14 | 2.36 |
| *Instruction-tuning experiments* | | | |
| MiniGPTv2 [9] | 6.78 | 10.56 | 6.94 |
| mPLUG-Owl3 [56] | 6.13 | 18.51 | 10.32 |
| InternVL-2.5 [10] | 18.11 | 29.17 | 18.28 |
| LLaVA-1.5 [32] | 8.59 | 16.81 | 9.79 |
| Qwen2.5-VL [6] | **23.64** | 33.39 | 24.50 |
| NAUTILUS(LLaVA-1.5) | 10.25 | 19.37 | 11.06 |
| NAUTILUS(Qwen2.5-VL) | 23.50 | **34.88** | **25.75** |

Table III: Comparisons on the image caption and region caption tasks.

| Method | Image Caption | | | Region Caption | | |
|---|---|---|---|---|---|---|
| | BLEU-4 ↑ | CIDEr ↑ | METEOR ↑ | BLEU-4 ↑ | CIDEr ↑ | METEOR ↑ |
| *Zero-shot experiments* | | | | | | |
| GPT-4o [40] | 0.047 | 0.322 | 0.179 | 0.018 | 0.137 | 0.148 |
| Qwen2.5-VL-72B [6] | 0.037 | 0.205 | 0.171 | 0.043 | 0.333 | 0.126 |
| Gemini 2.0 Flash [44] | 0.050 | 0.248 | 0.185 | 0.022 | 0.146 | 0.141 |
| *Instruction-tuning experiments* | | | | | | |
| MiniGPTv2 [9] | 0.107 | 0.794 | 0.204 | 0.097 | 0.792 | 0.178 |
| mPLUG-Owl3 [56] | 0.130 | 0.979 | 0.219 | **0.156** | 1.245 | **0.207** |
| InternVL-2.5 [10] | 0.126 | 0.924 | 0.208 | 0.147 | 1.177 | 0.195 |
| LLaVA-1.5 [32] | 0.126 | 0.902 | 0.208 | 0.141 | 1.104 | 0.189 |
| Qwen2.5-VL [6] | **0.140** | 1.015 | 0.222 | 0.149 | 1.230 | 0.196 |
| NAUTILUS(LLaVA-1.5) | 0.132 | 0.947 | 0.208 | 0.140 | 0.109 | 0.191 |
| NAUTILUS(Qwen2.5-VL) | 0.139 | **1.023** | **0.223** | 0.148 | **1.258** | 0.199 |

Table IV: Comparisons on the grounding task.

| Method | AP@0.5 ↑ | PR@0.75 ↑ | PR@0.5 ↑ |
|---|---|---|---|
| *Zero-shot experiments* | | | |
| GPT-4o [40] | 0.37 | 0.54 | 4.31 |
| Qwen2.5-VL-72B [6] | 30.90 | 33.11 | 46.36 |
| Gemini 2.0 Flash [44] | 6.42 | 7.55 | 20.60 |
| *Instruction-tuning experiments* | | | |
| MiniGPTv2 [9] | 32.75 | 41.19 | 50.99 |
| mPLUG-Owl3 [56] | 28.15 | 26.62 | 45.70 |
| InternVL-2.5 [10] | 37.50 | 44.69 | 54.64 |
| LLaVA-1.5 [32] | 31.04 | 30.46 | 48.21 |
| Qwen2.5-VL [6] | 39.34 | 47.42 | 57.62 |
| NAUTILUS(LLaVA-1.5) | 34.19 | 36.16 | 52.19 |
| NAUTILUS(Qwen2.5-VL) | **40.89** | **48.08** | **58.81** |

Table V: Comparisons on the coarse-grained and fine-grained classification tasks.

| Method | Coarse-grained | | | Fine-grained | | |
|---|---|---|---|---|---|---|
| | PR ↑ | F1 ↑ | acc ↑ | PR ↑ | F1 ↑ | acc ↑ |
| *Zero-shot experiments* | | | | | | |
| GPT-4o [40] | 84.40 | 63.46 | 55.18 | 67.58 | 43.43 | 54.44 |
| Qwen2.5-VL-72B [6] | 73.37 | 60.30 | 55.18 | 54.65 | 54.43 | 54.24 |
| Gemini 2.0 Flash [44] | 64.35 | 58.29 | 55.45 | 31.27 | 39.16 | 54.34 |
| *Instruction-tuning experiments* | | | | | | |
| MiniGPTv2 [9] | 82.17 | 79.53 | 79.95 | 89.94 | 89.91 | 90.00 |
| mPLUG-Owl3 [56] | **92.07** | **91.65** | **91.92** | 92.13 | 92.02 | 92.02 |
| InternVL-2.5 [10] | 91.01 | 90.79 | 91.25 | 90.34 | 90.33 | 90.40 |
| LLaVA-1.5 [32] | 90.11 | 89.46 | 90.04 | 89.72 | 89.72 | 89.80 |
| Qwen2.5-VL [6] | 82.31 | 82.94 | 85.33 | 90.02 | 88.19 | 88.18 |
| NAUTILUS(LLaVA-1.5) | 91.06 | 90.49 | 90.98 | 89.95 | 89.86 | 89.90 |
| NAUTILUS(Qwen2.5-VL) | 90.98 | 89.88 | 90.31 | **93.80** | **93.80** | **93.84** |

struggles to achieve optimal performance on the region caption task, second to mPLUG-Owl [56], a recent LMM renowned for its high-level image understanding performance, which also performs best on the coarse-grained classification task, demonstrated in Tab. V. Nonetheless, it is worth noting that the NAUTILUS presents impressive capabilities on the fine-grained classification task, surpassing other methods on three metrics. We attribute this to the effectiveness of the VFE module in enhancing the underwater vision feature, which empowers our models to perceive more fine-grained information.

## B.2 Ablation Study

We conduct ablation studies on the weighting strategies, with results presented in Tab. VI. In particular, we explore three designs of weighting strategies, including "wo/ weighting", "Learned from image feature", and "Learned from depth feature (ours)". Among them, the "wo/ weighting" strategy can be considered as "a norm weight 1". The "Learned from image feature" strategy, which means a loss of depth information, presents comparable performance compared with our design in most tasks. However, depth information is essential for distance perceptions of complex underwater objects, benefiting underwater scene understanding intuitively. Our strategy with the fusion of depth features surpasses other strategies, indicating the effectiveness of our design choice.

## B.3 Visualizations

We provide visualizations employing our NAUTILUS in more underwater scenes, as shown in Fig. II. Despite the significant diversity in these scenarios, including substantial variations in lighting

Table VI: Ablation studies on the weighting strategies. We employ Qwen2.5-VL [6] as the baseline model. Evaluations are conducted on eight tasks to provide a comprehensive analysis.

| Methods | Classification | | Caption | | Counting | Grounding | Detection | VQA |
|---|---|---|---|---|---|---|---|---|
| | Coarse | Fine | Image | Region | | | | |
| | acc ↑ | acc ↑ | METEOR ↑ | METEOR ↑ | MAE ↓ | PR@0.5 ↑ | AP@0.5 ↑ | METEOR ↑ |
| wo/ weighting | 87.9 | 89.1 | **0.222** | 0.197 | 36.8 | 55.4 | 35.9 | 0.367 |
| Learned from image feature | 88.6 | 90.2 | 0.220 | 0.195 | 36.5 | 55.7 | **36.3** | **0.375** |
| Learned from depth feature | **90.0** | **91.4** | 0.222 | 0.198 | **36.4** | **55.9** | 36.2 | 0.373 |

conditions and viewpoints, the NAUTILUS accurately localizes the targets and provides reasonable textual descriptions. These qualitative results further illustrate the effectiveness of our method.

## C  Discussion

The underwater imaging model leads a physics-driven research direction to tackle image degradation. Explicit information restoration in feature space presents a distinct advantage for our NAUTILUS, especially in high-level understanding tasks. Specifically, image augmentation changes pixel values, resulting in compromised preservation of the image's fidelity and semantics. In contrast, the proposed feature enhancement method incorporates physical priors to regularize the feature extraction and interaction process, preserving the original image information and demonstrating greater application potential.

As shown in Fig. III, the left and right columns show original and corresponding augmented underwater images. Feeding them into the representative multimodal large language model GPT-4o [40] produces markedly different feedback. In particular, while employing the data augmentation process, the GPT-4o misinterprets the *dim and diffuse* lighting as *nature and moderately bright*, illustrating the necessity of exploring feature enhancement in the underwater scene understanding task.

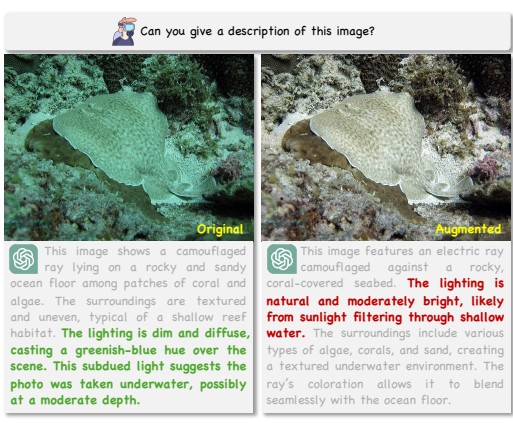

Figure III: Feeding a pair of original and augmented underwater images into an LMM yields output descriptions with notable differences, indicating that data augmentation can change the semantic interpretation of the image.

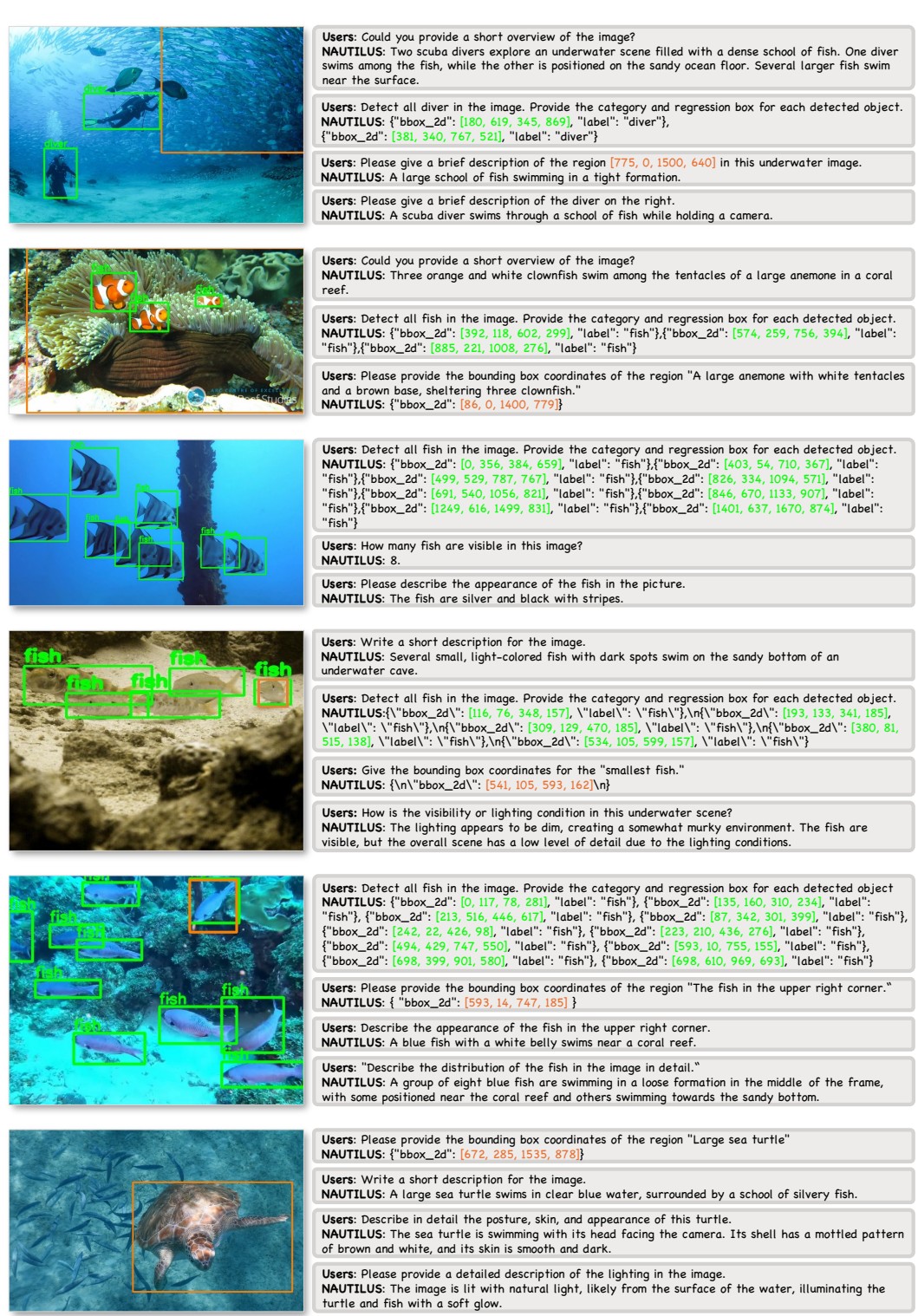

Figure II: Visualizations across various underwater scenes encompassing different illumination conditions, target categories, individual scales, environments, etc. Our NAUTILUS consistently presents remarkable underwater scene understanding performance.

Table VII: Prompts designed to generate text annotations.

**Prompt 1: Image caption**
You are an AI visual assistant analyzing an underwater image.
Task Overview:
Generate a detailed and accurate description of the image in one sentence.
Image caption guidelines:
1. Ensure that **each description is affirmative and can be inferred clearly from the image**.
2. Identify the main targets such as various fish, sea turtles, jellyfish and other marine organisms, shipwrecks and ruins, coral reefs, seagrass, divers, etc. When describing, mention the accurate number of targets and determine their category.
3. Consider the action or state of the objects, relationship between objects, background and environment.

**Prompt 2: Region caption**
Given an image, a bounding box (bbox), and additional textual context, generate a high-quality region description. The description must adhere to the following principles:
Input:
bounding box: {bbox}
This bounding box represents the normalized xycoordinates of the top-left and bottom-right corners of the target region in the image.
Accuracy: Ensure the description precisely reflects the content within the specified bbox without adding speculative or unrelated details.
Specificity: Provide concrete details about the object's attributes (e.g., shape, color, texture) and relevant contextual information.
Objectivity: Avoid any subjective interpretations, emotions, or assumptions about the object's purpose or intent.
Conciseness: Keep the description informative yet succinct, avoiding unnecessary elaboration.
Context Awareness: Consider the surrounding elements only if they are relevant to understanding the object in the bbox.
Output Format:
description: A darkcolored fish with a broad body and a slightly pointed head, swimming near the coral reef.

**Prompt 3: VQA**
You are an AI visual assistant analyzing an underwater image. Generate a structured dialogue between yourself and a person asking questions about the image. Your task is to create precise question-answer pairs based purely on the observable visual content of the image. Each answer should be as short as possible, preferably a single word or short phrase, while maintaining accuracy.
Task Overview:
Generate a variety of structured question-answer pairs that reflect the image's content. Questions should cover different aspects of the image and fall into one of the following categories: Object Recognition Questions: Identifying or detecting object types and categories (e.g., fish species, coral structures). Attribute Questions: Describing the properties of objects (e.g., color, size, shape, material). Counting Questions: Asking about the number of specific objects (e.g., number of fish or coral formations). Spatial Relation Questions: Asking about the relative position or spatial layout of objects (e.g., where objects are located or their relative positions ).
Guidelines:
For "Spatial Relation Questions",do not answer them by "In the ocean". Ensure that every question has a definite and clear answer based on what is visually observable in the image. Avoid speculative or ambiguous questions. Questions should be answerable with confidence and based on visible content. Include both simple (object identification, counting) and moderate (relative positioning, behaviors) questions.
Format:
Follow this exact format for each question-answer pair and no need to include other content:
Q: [Question]
A: [A single word or phrase]

Table VIII: A list of question templates employed to construct task-specific conversions.

**Task 1: Image caption**
1. Write a description for the image.
2. Offer a concise description of the image.
3. Give a description of the scene.
4. Give a short description of the image.
5. Provide a brief description of the image.
6. Please give a succinct description of what is shown in this image.
7. Could you describe the image briefly?
8. Could you provide a description of the image?
9. Can you give a brief description of this image?
10. Could you provide a short overview of the image?

**Task 2: Region caption**
1. Please provide a concise description of this region [bbox] in this underwater image.
2. Please give a brief description of the region [bbox] in this underwater image.
3. Could you provide a concise description of the region [bbox] in this underwater image?
4. Please offer a succinct description of the region [bbox].
5. Please provide a short yet informative description of the region [bbox].
6. Describe this region [bbox] in the underwater image.
7. Briefly describe the content of this region [bbox].
8. In this underwater image, please provide a concise description of the region [bbox].
9. For this underwater image, concisely describe the content of the region [bbox].
10. What's in this region [bbox] of the underwater image? Describe it concisely.

**Task 3: Grounding**
1. Please locate the bounding box coordinates of the [region].
2. Find and return the bounding box coordinates of the [region].
3. Give the bounding box coordinates for the [region].
4. Extract the precise bounding box coordinates that correspond to the given [region].
5. Detect and outline the bounding box coordinates enclosing the [region].

**Task 4: Detection**
1. Detect all [class] object in the image.
2. Detect all underwater object in the image, including [class1], [class2], ..., [classn].

**Task 5: Counting**
1. How many fish can you find in this image?
2. Please count the fish in the image.
3. Identify the number of fish present in this image.
4. Count the total number of fish visible in this image.
5. What is the count of fish in this image?
6. How many fish can you see in this image?
7. How many fish are visible in this image?
8. Please count how many fish are in this image.
9. Can you determine the number of fish in this image?
10. What is the total number of fish shown in this image?

**Task 6: Coarse-grained classification**
1. Identify the object inside the specified regression box [bbox].
2. Categorize the object located within the provided regression box [bbox] in the underwater image.
3. Classify the items found inside the given regression box [bbox].
4. Determine the category of the object inside the specified regression box [bbox].
5. Assign a category to the object inside the provided regression box [bbox] in the image.

**Task 7: Fine-grained classification**
1. Please identify the biological class of fish depicted in the image.
2. Can you recognize the fish taxonomic class shown in this image?
3. Could you determine the taxonomic class of the fish in the image?
4. What is the biological class of the fish in the image?
5. You are requested to identify the biological class of fish present in the image.

