# OpenReview forum: "NAUTILUS: A Large Multimodal Model for Underwater Scene Understanding"
_NeurIPS.cc/2025/Conference — NeurIPS 2025 poster_

### Official Review · Reviewer_Aru7 · 2025-06-30

**Clarity:** 4
**Significance:** 2
**Originality:** 2
**Rating:** 4
**Confidence:** 3

**Summary:**

This paper presents NAUTILUS, a large vision-language model tailored for underwater scene understanding. To tackle the challenges of underwater image degradation and the lack of diverse datasets, the authors introduce NautData—a new instruction-tuning dataset with 1.45 million image-text pairs spanning eight tasks across image, region, and object levels. They also propose a plug-and-play Vision Feature Enhancement (VFE) module based on physical underwater imaging principles, which improves feature quality by correcting for light scattering and absorption. Integrated into strong LMM backbones like LLaVA-1.5 and Qwen2.5-VL, NAUTILUS achieves state-of-the-art performance across a range of benchmarks, demonstrating robust and comprehensive underwater perception capabilities.

**Questions:**

- While the model shows strong performance on NautData and select public benchmarks, it’s unclear how well NAUTILUS generalizes to truly unseen underwater environments or rare object categories not covered in the training data. Have the authors considered evaluating on additional real-world or long-tail settings to test robustness? Results or discussion in this direction would strengthen the claims on general applicability.
- Since much of NautData is constructed using LMMs like Gemini, Qwen, and GPT-4o, it raises concerns about annotation accuracy—especially given that Table 2 shows GPT-4o itself performs poorly on the benchmark. Could the authors clarify how they ensured label correctness during dataset construction? Additionally, quantitative measures of annotation quality (e.g., human agreement rates or error analyses) would help justify the dataset’s reliability as a benchmark.

**Ethical Concerns:**

["NO or VERY MINOR ethics concerns only"]

**Final Justification:**

Most of my concerns have been addressed in the rebuttal and discussions. I am satisfied with the technical quality and clarity of the work, and will maintain my current score. My only remaining concern is the limited scope of contributions, which may affect the broader impact of the paper. I remain open to either outcome depending on how this aspect is weighed by the AC and other reviewers.

**Limitations:**

yes

**Quality:**

3

**Strengths And Weaknesses:**

**Strengths**

1. The paper introduces a high-quality dataset, NautData, which is large, diverse, and well-annotated across eight underwater tasks. This is a meaningful contribution to a relatively underexplored domain.
2. The Vision Feature Enhancement (VFE) module leverages physical priors to address underwater image degradation at the feature level.
3. The paper is well-written and easy to follow.

---

**Weaknesses**

- While the proposed VFE module is grounded in a physical imaging model, the paper does not clearly define the applicability limits of the model, such as how well it generalizes under extreme lighting or turbidity conditions. The robustness of the dark-pixel prior is also not thoroughly analyzed—its reliability may vary depending on scene composition and noise.
- In addition, the scope of contributions is relatively narrow, focusing primarily on the construction of a dataset and the design of the VFE module. While both are valuable, the paper would benefit from broader methodological innovations or deeper analysis of the interplay between physical priors and large-scale vision-language models.

---

> ### Author Rebuttal · Authors · 2025-07-31
>
> Thanks for providing feedback and taking the time to review our work!
>
> - **To Weakness 1**: “While the proposed VFE module is grounded in a physical imaging model, the paper does not clearly define the applicability limits of the model, such as how well it generalizes under extreme lighting or turbidity conditions. The robustness of the dark-pixel prior is also not thoroughly analyzed—its reliability may vary depending on scene composition and noise.”
>
> **Reply:** Thanks. Following your advice, we evaluate the performance of our models under various lighting and turbidity conditions. Specifically, we employ Gemini 2.5 Flash to filter for images of low-light, normal-light, turbid, and clear from the NautData test set. The low-light and turbid images suffer from underwater degradations with extreme lighting and turbidity conditions, respectively. We conduct comparisons separately on the low-light, normal-light, turbid, and clear subsets, with results presented in the table below. It is worth noting that the VFE module brings remarkable performance gains, especially under the low-light and turbid conditions for up to +5.2 and +2.8 PR\@0.5 improvements, respectively, demonstrating the great generalization capabilities of the VFE module.
>
> | Method | Low-light | Normal-light | Turbid | Clear |
> | --- | --- | --- | --- | --- |
> | MiniGPTv2 | 26.6 | 42.0 | 34.7 | 41.1 |
> | mPLUG-Owl3 | 28.3 | 35.9 | 27.3 | 38.8 |
> | InternVL-2.5 | 29.1 | 42.6 | 36.7 | 40.9 |
> | LLaVA-1.5 | 26.2 | 35.7 | 28.8 | 36.8 |
> | **NAUTILUS**(LLaVA-1.5) | 27.7(+1.5) | 37.6(+1.9) | 31.6(+2.8) | 37.9(+1.1) |
> | Qwen2.5-VL | 30.6 | 43.6 | 38.0 | 42.3 |
> | **NAUTILUS**(Qwen2.5-VL) | 35.8(+5.2) | 44.4(+0.8) | 39.6(+1.6) | 44.0(+1.7) |
>
> Subsequently, we evaluate the robustness of the dark-pixel prior. As shown in the following table, randomly selected dark pixels lead to a significant performance decline of up to 16.1 MAE and 13.0 AP\@0.5 compared with adopting the dark-pixel prior, illustrating the effectiveness of the dark-pixel prior in most scenarios. We will add analysis and discussions in the revised version.
>
> | Strategy of sampling dark pixel | Coarse-grained Classification | Fine-grained Classification | Image Caption | Region Caption | Counting | Grounding | Detection | VQA |
> | --- | --- | --- | --- | --- | --- | --- | --- | --- |
> |  | acc ↑ | acc ↑ | meteor ↑ | meteor ↑ | MAE ↓ | PR\@0.5 ↑ | AP\@0.5 ↑ | meteor ↑ |
> | random pixel | 87.7 | 86.2 | 0.201 | 0.185 | 51.4 | 36.4 | 31.8 | 0.330 |
> | dark-pixel prior | 93.1 | 92.3 | 0.221 | 0.189 | 35.3 | 40.3 | 44.8 | 0.354 |
>
> - **To Weakness 2**: “In addition, the scope of contributions is relatively narrow, focusing primarily on the construction of a dataset and the design of the VFE module. While both are valuable, the paper would benefit from broader methodological innovations or deeper analysis of the interplay between physical priors and large-scale vision-language models.”
>
> **Reply:** Thanks for admiring the value of our work! Underwater understanding enables automated explorations into the most extensive scope of our planet, facilitating applications in resource exploration, robot navigation, etc. However, the absence of multi-granular underwater LMMs hinders the deep development of this area. We take a pioneering step towards underwater LMMs from dataset construction and model design aspects, contributions we believe are valuable foundations for subsequent research.
>
> Physical priors present an interpretable way to construct LMMs. In particular, the underwater imaging model inspires us to take advantage of depth modality while overcoming the underwater degradation. Specifically, the underwater imaging model highlights light absorption during underwater propagation, challenging the quality of underwater images, especially the targets far away from the cameras. Thus, we integrate depth encoder in the design of our VFE module, regularizing feature fusion of depth and vision modalities in a reasonable framework.
>
> Additionally, we provide visualizations to give an intuitive impression of the VFE module’s effectiveness. Due to the policy restrictions during the NeurIPS rebuttal phase, we cannot directly present figures. Thus, we upload these visualizations as Fig.1 via the anonymous link embedded in the supplementary file (this does not violate the rebuttal requirements). We sincerely request that the reviewer click this anonymous link for more details. The visualizations demonstrate that the original vision features struggle with focusing on objects of interest, while the enhanced features present high responses to them, illustrating remarkable potential to achieve reliable underwater scene understanding. We will add these discussions to the revised version.
>
>
> - **To Question 1**: “While the model shows strong performance on NautData and select public benchmarks, it’s unclear how well NAUTILUS generalizes to truly unseen underwater environments or rare object categories not covered in the training data. Have the authors considered evaluating on additional real-world or long-tail settings to test robustness? Results or discussion in this direction would strengthen the claims on general applicability.”
>
> **Reply:** Thanks. We have evaluated zero-shot performance on the Marine-Inst20M [3], an unseen real-world dataset. Our NAUTILUS surpasses other methods on this dataset, presenting a strong generalization performance. Following your advice, we add zero-shot evaluations on UIIS10K [1] and TrashCan [2], two additional datasets not involved in the training data. In particular, the UIIS10K covers rare categories such as “Artiodactyla” and “Mollusk,” and the TrashCan is captured in entirely unseen underwater environments. Both of them are real-world datasets and provide bounding box annotations.
> We sample 200 images from each of the two datasets in our experiments for evaluation. Specifically, we employ Gemini 2.5 Flash to generate descriptions for annotated instances and manually verify each sentence by our team to ensure the accuracy of the data quality. Subsequently, we conduct zero-shot evaluations on these data, with performances recorded in the following table. Our NAUTILUS surpasses the counterparts by at least 3.0 PR\@0.5 and 1.5 PR\@0.5 on the UIIS10K and TrashCan datasets, respectively, presenting a remarkable performance of general applicability. We will add these evaluations in the revised version.
>
> | **Method** | **UIIS10K** [1] | **TrashCan** [2] | Marine-Inst20M [3] |
> | --- | --- | --- | --- |
> | MiniGPTv2 | 43.5 | 19.5 | 37.3 |
> | mPLUG-Owl3 | 53.5 | 31.5 | 36.0 |
> | InternVL-2.5 | 56.0 | 41.7 | 38.3 |
> | LLaVA-1.5 | 51.5 | 24.5 | 36.0 |
> | **NAUTILUS**(LLaVA-1.5) | 53.0(+1.5) | 29.0(+4.5) | 37.1(+1.1) |
> | Qwen2.5-VL | 62.5 | 47.5 | 40.0 |
> | **NAUTILUS**(Qwen2.5-VL) | 65.5(+3.0) | 49.0(+1.5) | 40.4(+0.4) |
>
> [1] UWSAM: Segment Anything Model Guided Underwater Instance Segmentation and A Large-scale Benchmark Dataset. arXiv 25.
>
> [2] TrashCan: A Semantically-Segmented Dataset towards Visual Detection of Marine Debris. arXiv 20.
>
> [3] MarineInst: A Foundation Model for Marine Image Analysis with Instance Visual Description. ECCV 24.
>
> - **To Question 2**: “Since much of NautData is constructed using LMMs like Gemini, Qwen, and GPT-4o, it raises concerns about annotation accuracy—especially given that Table 2 shows GPT-4o itself performs poorly on the benchmark. Could the authors clarify how they ensured label correctness during dataset construction? Additionally, quantitative measures of annotation quality (e.g., human agreement rates or error analyses) would help justify the dataset’s reliability as a benchmark.”
>
> **Reply:** The NautData can be divided into three parts according to their generation method. Specifically, data for counting, detection, coarse-grained, and fine-grained classification tasks are generated rule-based, without involving LMMs. Annotations of this part of the data are obviously accurate. Indeed, the performance of GPT-4o is significantly inferior to that of other models. However, the GPT-4o is not employed to generate bounding box annotations and is only used to filter out low-quality text descriptions, unleashing the superior multi-modality performance of GPT-4o. Besides, employing LMMs in data scaling of large-scale instruction-following datasets is a prevailing practice [1][2], which introduces noise while maintaining the overall accuracy.
>
> To evaluate the label correctness of our NautData, following your advice, we conduct human studies, providing a quantitative measure of annotation quality. Specifically, we adopt a ten-point scoring scale. For example, score 10 denotes “The description is exhaustive, internally consistent, and entirely accurate; no factual, logical, or interpretive errors are observed”, score 8 denotes “Generally accurate, but contains several fine-grained errors that exert limited impact on core understanding”, score 3 denotes “Poor: Predominantly inaccurate; only a few statements are correct, and the overall presentation is misleading”, and score 1 denotes “The content is entirely wrong, irrelevant, or contradicts established knowledge; no redeeming factual accuracy is present”. Fine-grained errors like misinterpreting “clownfish” as “acrossocheilus” demand a score of eight, while misinterpreting “fish” as “crab” will just obtain a score of six. Our dataset finally obtains an average score of 8.06, illustrating an overall high quality of the NautData with tolerable noise. We will add discussions of our data quality in the revised version.
>
> [1] ShareGPT4V: Improving Large Multi-modal Models with Better Captions. ECCV 24.
>
> [2] GROUNDHOG: Grounding Large Language Models to Holistic Segmentation. CVPR 24.

---

> > ### Comment · Reviewer_Aru7 · 2025-08-06
> >
> > Most of my concerns have been addressed. I will maintain my current score.

---

> > > ### Author Response · Authors · 2025-08-06
> > >
> > > Dear Reviewer Aru7,
> > >
> > > Thank you for your time and kind feedback. Your comments are valuable to us in improving the quality of this work. We will incorporate your suggestions in the revision. Many thanks again!
> > >
> > > Best regards,
> > >
> > > Paper 3623 Authors

---

### Official Review · Reviewer_qKHH · 2025-07-01

**Clarity:** 3
**Significance:** 2
**Originality:** 3
**Rating:** 4
**Confidence:** 3

**Summary:**

This work makes two primary contributions to underwater scene understanding:

(1) the introduction of NautData, a novel large-scale multimodal dataset encompassing 1.45M image-text pairs across eight tasks spanning image-, region-, and object-level understanding; and

(2) NAUTILUS, a new underwater-adapted LMM that integrates a physics-inspired vision feature enhancement (VFE) module to
explicitly mitigate degradation effects in underwater imagery.

The proposed VFE module demonstrates strong plug-and-play compatibility when incorporated into established LMMs (LLaVA-1.5, Qwen-VL), with experiments showing consistent performance gains across multiple underwater understanding tasks.
By addressing both the data scarcity challenge through NautData and the degradation robustness challenge through the VFE design,
this work provides a valuable foundation for future research in underwater multimodal perception systems.

**Questions:**

(1) What motivated the choice of feature-space enhancement over traditional image-space approaches?
Are there specific underwater degradation characteristics that make this more effective?

(2) Could you provide either: A more rigorous theoretical proof connecting VFE's operations to underwater imaging models,
or visualizations showing how VFE module removes underwater degradation?

(3) Could you add experiments isolating the contribution of the dark pixel prior in VFE and performance comparison between feature-space and image-space enhancement?

(4) In the 4th row of Supplementary Figure II, NAUTILUS incorrectly localizes the leftmost cuttlefish. Could you explain why?

My evaluation scores could be positively revised should the authors substantively address Q1-Q3 with new analyses and discuss the Figure II anomaly's implications.

**Ethical Concerns:**

["NO or VERY MINOR ethics concerns only"]

**Final Justification:**

Most of my concerns have been addressed in the rebuttal and discussions. The additional experiments provided by the authors during the rebuttal phase effectively address the limitations in the paper. Incorporating these experimental results and other visualizations in the final version would significantly enhance the completeness and effectiveness of the article. I am upgrading my score to 4 (borderline accept).

**Limitations:**

yes

**Paper Formatting Concerns:**

I didn't notice any major formatting issues in this paper.

**Quality:**

3

**Strengths And Weaknesses:**

Strengths:

(1) The construction of NautData represents a significant contribution, being the first large-scale dataset to
comprehensively cover eight underwater understanding tasks across image-, region-, and object-level granularities.

(2) The proposed NAUTILUS model demonstrates consistent and measurable improvements across multiple underwater tasks when
benchmarked against established baselines.

(3) The paper is well-written with clear organization and minimal grammatical errors, facilitating efficient comprehension of the technical content.


Weaknesses:

(1) This paper lacks explanation for performing enhancement in feature space rather than image space

(2) This paper lacks rigorous mathematical proof establishing the connection between the VFE module's design principles and classical underwater imaging physics

(3) Critical components of the VFE design lack empirical validation, including the impact of feature-space vs. pixel-space enhancement and the choise of the dark pixel prior.

---

> ### Author Rebuttal · Authors · 2025-07-31
>
> Thanks for providing feedback and taking the time to review our work!
>
> - **To Weakness 1 & Question 1**: “… lacks explanation for performing enhancement in feature space rather than image space” / “What motivated the choice of feature-space enhancement over traditional image-space approaches? Are there specific underwater degradation characteristics that make this more effective?”
>
> **Reply:** Thanks. There are three observations motivating us to explore feature-space enhancement over image-space approaches:
>
> 1) Traditional image restoration methods align with human visual quality preferences. However, “the factors contributing to **visual quality often differ from those determining recognition quality**” **[1][2]**, while the high recognition quality is our purpose.
>
> 2) Image-space restoration methods directly **alter the raw signal**, hurting the original image information. Instead, the feature-space enhancement completely preserves the input of original pixel values, enhancing underwater information more softly.
>
> 3) Decoupling image enhancement from model training can accumulate errors. This **risk is magnified in our large-scale underwater dataset** captured under highly variable conditions. Enhancement in feature space enables end-to-end optimization, allowing task-oriented supervision from downstream tasks, which could be more suitable for our eight-task understanding models.
>
> We pursue performance improvement on underwater understanding rather than prioritizing human-oriented visual quality, and therefore, we investigate feature-space enhancement approaches. Besides, we provide empirical validations to evaluate the effectiveness of feature-space enhancement. As shown in the following table, our method improves the counting, grounding, and detection performance by at least 2.7 points, demonstrating a remarkable potential to explore a feature-space enhancement approach. We will add these discussions to the revised version.
>
> | Restoration Methods | Coarse-grained Classification | Fine-grained Classification | Image Caption | Region Caption | Counting | Grounding | Detection | VQA |
> | --- | --- | --- | --- | --- | --- | --- | --- | --- |
> |  | acc ↑ | acc ↑ | meteor ↑ | meteor ↑ | MAE ↓ | PR\@0.5 ↑ | AP\@0.5 ↑ | meteor ↑ |
> | Qwen2.5-VL(Baseline) | 89.1 | 92.1 | 0.220 | **0.189** | 40.7 | 37.6 | 41.9 | **0.357** |
> | +Reti-Diff [3] | 92.3 | 91.7 | 0.217 | **0.189** | **33.7** | 38.1 | 36.3 | 0.355 |
> | +SMDR-IS [4] | 88.3 | 85.8 | 0.214 | 0.182 | 38.9 | 39.9 | 32.4 | 0.352 |
> | +CCL-Net [5] | 89.7 | 88.6 | 0.217 | 0.183 | 45.7 | 36.9 | 38.5 | 0.353 |
> | +VFE (ours) | **93.1** | **92.3** | **0.221** | **0.189** | 35.3 | **40.3** | **44.8** | 0.354 |
>
> [1] UniRestore: Unified Perceptual and Task-Oriented Image Restoration Model Using Diffusion Prior. CVPR 25.
>
> [2] Visual Recognition-Driven Image Restoration for Multiple Degradation with Intrinsic Semantics Recovery. CVPR 23.
>
> [3] Reti-Diff: Illumination Degradation Image Restoration with Retinex-based Latent Diffusion Model. ICLR 25.
>
> [4] Synergistic multiscale detail refinement via intrinsic supervision for underwater image enhancement. AAAI 24.
>
> [5] Underwater Image Enhancement With Cascaded Contrastive Learning. TMM 24.
>
> - **To Weakness 2 & Question 2**: “...establishing the connection between the VFE module's design principles…”/ “… visualizations showing how VFE module removes underwater degradation?”
>
> **Reply:** Thanks for this insightful comment! The VFE module consists of two components motivated by the physical underwater imaging model: 1) employing dark pixels to localize the backscattering response, and 2) introducing depth features to estimate the restoration weight, alleviating information degradation during underwater propagation. Method in [1] provides strategies to estimate backscattering response and restoration weight within the physical underwater imaging model, and achieves underwater image restoration. Taking inspiration from this method, we revive its idea in feature space, first estimating representations of unavailable items in the physical underwater imaging model and subsequently substituting them into the equation of the underwater imaging physics to obtain an enhanced vision feature.
>
> Besides, following your advice, to intuitively illustrate the effectiveness of our VFE module, we provide visualizations of the original and enhanced vision features. Due to the policy restrictions during the NeurIPS rebuttal phase, we are unable to directly present figures. Thus, we upload these visualizations as Fig.1 **via the anonymous link embedded in the supplementary file** (this does not violate the rebuttal requirements).
>
> We can see that the original feature struggles to focus on the targets of interest while the enhanced features clearly find them with high responses, demonstrating the remarkable performance of the VFE model in understanding underwater scenes. We will meticulously refine the relevant discussions and analysis in the revised version.
>
> [1] What is the Space of Attenuation Coefficients in Underwater Computer Vision? CVPR 17.
>
> [2] UWSAM: Segment Anything Model Guided Underwater Instance Segmentation and A Large-scale Benchmark Dataset. arXiv 25.
>
> - **To Weakness 3 & Question 3**: “Critical components of the VFE design lack empirical validation, including the impact of feature-space vs. pixel-space enhancement and the choise of the dark pixel prior.” / “Could you add experiments isolating the contribution of the dark pixel prior in VFE and performance comparison between feature-space and image-space enhancement?”
>
> **Reply:** Thanks. Following your advice, we conduct empirical validation to evaluate the critical components of the VFE design. First, we compare the effectiveness of feature-space and pixel-space enhancements. As shown in the table below, SMDR-IS and CCL-Net introduce a performance decline on most tasks. Reti-Diff suffers a 5.6 AP\@0.5 performance drop on the detection task. We attribute this to challenges from complex underwater scenes and information loss during pixel-space enhancement. In contrast, our VFE module preserves the original information, achieving 4.0 acc, 5.4 MAE, 2.7 PR\@0.5, and 2.9 AP\@0.5 improvements on the coarse-grained classification, counting, grounding, and detection tasks.
>
> | Restoration Methods | Coarse-grained Classification | Fine-grained Classification | Image Caption | Region Caption | Counting | Grounding | Detection | VQA |
> | --- | --- | --- | --- | --- | --- | --- | --- | --- |
> |  | acc ↑ | acc ↑ | meteor ↑ | meteor ↑ | MAE ↓ | PR\@0.5 ↑ | AP\@0.5 ↑ | meteor ↑ |
> | Qwen2.5-VL(Baseline) | 89.1 | 92.1 | 0.220 | 0.189 | 40.7 | 37.6 | 41.9 | 0.357 |
> | +Reti-Diff [1] | 92.3 | 91.7 | 0.217 | 0.189 | 33.7 | 38.1 | 36.3 | 0.355 |
> | +SMDR-IS [2] | 88.3 | 85.8 | 0.214 | 0.182 | 38.9 | 39.9 | 32.4 | 0.352 |
> | +CCL-Net [3] | 89.7 | 88.6 | 0.217 | 0.183 | 45.7 | 36.9 | 38.5 | 0.353 |
> | +VFE (ours) | 93.1 | 92.3 | 0.221 | 0.189 | 35.3 | 40.3 | 44.8 | 0.354 |
>
> Subsequently, we evaluate the choice and contribution of the dark pixel prior. In particular, treating the design of “wo/ dark prior” as the baseline, randomly sampled dark pixels may misinterpret clear responses as backscattering, destroying original semantic information, and suffering a significant performance decline of 12.1 AP\@0.5 on the detection task. Afterwards, we design a strategy to sample the image patch containing the most pixels with the minimum RGB values. However, we also observe a decline in performance on six of eight takes, which we attribute to the noises involved in underwater environments containing pixels with meaningless artifacts. Therefore, we compute mean RGB values to mitigate the interference of noise and achieve remarkable improvements, illustrating the contribution of the dark pixel prior in our method. We will add these validations and discussions in the revised version.
>
> | Choice of dark prior | Coarse-grained Classification | Fine-grained Classification | Image Caption | Region Caption | Counting | Grounding | Detection | VQA |
> | --- | --- | --- | --- | --- | --- | --- | --- | --- |
> |  | acc ↑ | acc ↑ | meteor ↑ | meteor ↑ | MAE ↓ | PR\@0.5 ↑ | AP\@0.5 ↑ | meteor ↑ |
> | wo/ dark prior | 92.7 | 91.3 | 0.217 | 0.191 | 36.5 | 38.5 | 43.9 | 0.356 |
> | random pixel | 87.7 | 86.2 | 0.201 | 0.185 | 51.4 | 36.4 | 31.8 | 0.330 |
> | containing most pixels with minimum RGB value | 89.6 | 88.5 | 0.209 | 0.192 | 45.6 | 37.6 | 38.2 | 0.341 |
> | minimum of mean RGB value (ours) | 93.1 | 92.3 | 0.221 | 0.189 | 35.3 | 40.3 | 44.8 | 0.354 |
>
> [1] Reti-Diff: Illumination Degradation Image Restoration with Retinex-based Latent Diffusion Model. ICLR 25.
>
> [2] Synergistic multiscale detail refinement via intrinsic supervision for underwater image enhancement. AAAI 24.
>
> [3] Underwater Image Enhancement With Cascaded Contrastive Learning. TMM 24.
>
> - **To Question 4**: “ In the 4th row of Supplementary Figure II, NAUTILUS incorrectly localizes the leftmost cuttlefish. Could you explain why?”
>
> **Reply:** Thank you for carefully reading this paper. Although our NAUTILUS exhibits remarkable underwater scene understanding performance, there are still challenging cases that occasionally raise errors. The 4th row of Supplementary Figure II is a failure case for the grounding task. It is challenging for a group of fish with the same fine-grained categories represented in clusters, requiring the method to distinguish the detailed instance appearance from the image modality and the delicate position descriptions from the text modality. We study to address these failure cases in future work.
>
> - My evaluation scores could be positively revised should the authors substantively address Q1-Q3 with new analyses and discuss the Figure II anomaly's implications.
>
> **Reply:** Thanks for your insightful comments. We hope we have already addressed your concerns. If you have any further questions, please feel free to let us know.

---

> ### Author Response · Authors · 2025-08-04
> **We are open to any further discussion.**
>
> Dear Reviewer qKHH,
>
> We sincerely appreciate your time and effort in reviewing our paper. We hope our explanations have addressed your concerns. As the discussion phase is nearing its end, we look forward to your reply. If further clarification is needed, please do not hesitate to mention it, and we will promptly address your inquiries. We thank you once again for your time.
>
> Best regards,
>
> Paper 3623 Authors

---

> > ### Comment · Reviewer_qKHH · 2025-08-05
> > **Official Comment by Reviewer qkHH**
> >
> > Dear Authors of Paper 3623,
> >
> > Thank you for your comprehensive rebuttal. Your detailed responses have resolved my initial concerns regarding the paper. I appreciate the clarity of your arguments.
> >
> > Sincerely,
> > Reviewer qKHH

---

> > > ### Author Response · Authors · 2025-08-06
> > >
> > > Dear Reviewer qKHH,
> > >
> > > Thank you for your constructive comments. We will incorporate your suggestions and the above discussions in the next version. We would greatly appreciate it if you could consider improving the evaluation after reviewing our responses. Thank you very much for your consideration.
> > >
> > > Best regards,
> > >
> > > Paper 3623 Authors

---

### Official Review · Reviewer_R364 · 2025-07-02

**Clarity:** 3
**Significance:** 3
**Originality:** 3
**Rating:** 4
**Confidence:** 3

**Summary:**

This study addresses the challenges of data scarcity and poor image quality in underwater scene understanding by proposing a large vision-language model named NAUTILUS. To achieve this, the researchers first constructed a large-scale multi-task instruction dataset for underwater environments, NautData, containing 1.45 million image-text pairs. They then designed an innovative plug-and-play Visual Feature Enhancement (VFE) module, which leverages the physical principles of underwater imaging to effectively mitigate image degradation.

**Questions:**

Could the authors analyze the noise levels and perform error analysis?

**Ethical Concerns:**

["NO or VERY MINOR ethics concerns only"]

**Final Justification:**

The paper proposed the large vision-language model for underwater tasks, trained on the a large scale dataset and enhanced by a physics-inspired VFE module, achieving superior performance over GPT-4o and Gemini 2.0 across most tasks. Most of my concerns have been addressed during the rebuttal phase thus I choose to keep my rating.

**Limitations:**

Yes

**Quality:**

3

**Strengths And Weaknesses:**

Strengths:
- The researchers developed NautData, a large-scale underwater instruction learning dataset comprising 1.45 million question-answer pairs. This dataset supports eight different tasks.
- The study introduces a plug-and-play Visual Feature Enhancement (VFE) module inspired by the physical principles of underwater imaging.
- NAUTILUS is the first large vision-language model capable of handling eight underwater tasks.
- Experiments show that NAUTILUS outperforms models including GPT-4o and Gemini 2.0 on most tasks in the NautData test set.

Weaknesses:
- The paper does not analyze the noise level within the LLM-generated portion of the NautData dataset, nor does it discuss how this potential noise might affect the model's training and final performance.
- The research lacks a qualitative analysis of failure cases. Discussing specific examples where the model fails would offer valuable insights into its limitations and areas for future improvement.
- The reported performance improvements over the LLaVA-1.5 baseline are quite marginal on several tasks. This limited gain could suggest that the proposed method's effectiveness and generalization capabilities may not be as significant.
- The study would be strengthened by an ablation that compares this method to other, more naive weighting strategies to justify its design choice.

---

> ### Author Rebuttal · Authors · 2025-07-31
>
> Thank you for your valuable feedback!
>
> - **To Weakness 1 & Question 1**: “The paper does not analyze the noise level within the LLM-generated portion of the NautData dataset, nor does it discuss how this potential noise might affect the model's training and final performance.” / “Could the authors analyze the noise levels and perform error analysis?”
>
> **Reply:** Thanks. Hallucinations in LLMs are a fundamental issue that is difficult to avoid entirely [1][2][3], introducing noises into the construction of large-scale instruction-tuning datasets. The demand for data scaling makes it impractical for us to assess the quality of each image-text pair by humans. Thus, to evaluate the quality of our NautData, we randomly select two hundred image-text samples and recruit four volunteers at the master’s level to give their scores on the quality of each sample.
>
> Specifically, we adopt a ten-point scoring scale. For example, score 10 denotes “The description is exhaustive, internally consistent, and entirely accurate; no factual, logical, or interpretive errors are observed”, score 8 denotes “Generally accurate, but contains several fine-grained errors that exert limited impact on core understanding”, score 3 denotes “Poor: Predominantly inaccurate; only a few statements are correct, and the overall presentation is misleading”, and score 1 denotes “The content is entirely wrong, irrelevant, or contradicts established knowledge; no redeeming factual accuracy is present”. Fine-grained errors like misinterpreting “clownfish” as “acrossocheilus” demand a score of eight, while misinterpreting “fish” as “crab” will just obtain a score of six.
>
> The final average score of our dataset is 8.06, illustrating an overall high quality of the NautData with tolerable hallucinations. We attribute this final assessment of scoring 8.06 to two reasons: 1) four of the eight supported tasks, including the coarse-grained, fine-grained, grounding, and detection tasks, are manually annotated and thus free of hallucinatory noise, and 2) we integrate LMM filtering followed by manual correction, which is a practical approach to minimize dataset noise.
>
> Besides, training our models on the 1/3 portion and the entire portion of the NautData, we observe an obvious scaling law of our dataset for 1.5 acc, 1.8 acc, 3.2 MAE, 2.0 PR\@0.5, and 2.8 AP\@0.5 improvements on the coarse-grained, fine-grained, counting, grounding, and detection tasks, respectively. This finding demonstrates that the advantages of data scaling still outweigh the detrimental effects of the noise present in our dataset. We will add these analyses of data quality in the revised version.
>
> | Training data | Coarse-grainedClassification | Fine-grainedClassification | Counting | Grounding | Detection |
> | --- | --- | --- | --- | --- | --- |
> |  | acc ↑ | acc ↑ | MAE ↓ | PR\@0.5 ↑ | AP\@0.5 ↑ |
> | 1/3 of NautData | 93.1 | 92.3 | 35.3 | 40.3 | 44.8 |
> | Entire NautData | 94.6(+1.5) | 94.1(+1.8) | 32.1(+3.2) | 42.3(+2.0) | 47.6(+2.8) |
>
> [1] Hallucination Augmented Contrastive Learning for Multimodal Large Language Model. CVPR 24.
>
> [2] LLM-Check: Investigating Detection of Hallucinations in Large Language Models. NeurIPS 24.
>
> [3] A Survey on Hallucination in Large Language Models: Principles, Taxonomy, Challenges, and Open Questions. ACM TIS 25.
>
> - **To Weakness 2**: “The research lacks a qualitative analysis of failure cases. Discussing specific examples where the model fails would offer valuable insights into its limitations and areas for future improvement.”
>
> **Reply:** Good suggestion! We conduct qualitative analysis on the NautData test set and observe failure cases in extremely dense scenes, i.e., the counting error could increase to ten percent while thousands of fish swim in groups. Another cases indicate that while instances of the same fine-grained category occur in adjacent clusters, grounding errors occasionally arise. We will add these analyses with corresponding qualitative results in the revised version.
>
> - **To Weakness 3**: “The reported performance improvements over the LLaVA-1.5 baseline are quite marginal on several tasks. This limited gain could suggest that the proposed method's effectiveness and generalization capabilities may not be as significant.”
>
> **Reply:** Thanks. We provide exhaustive evaluations on this phenomenon, analyzing the factors contributing to LLaVA-1.5’s performance improvements. Specifically, we divide the NautData test set into subsets characterized by low-light, normal-light, green-color, and blue-color conditions. Subsequently, we separately evaluate our models on these subsets and observe a notable improvement of +4.8 PR\@0.5 under the green-color condition, presenting a remarkable effectiveness for handling challenging underwater degradations. Therefore, we will provide fine-grained subsets of our NautData to better illustrate the advantages of our methods.
>
> | **Method** | **Low-light** | **Normal-light** | **Green-color** | **Blue-color** |
> | --- | --- | --- | --- | --- |
> | MiniGPTv2 | 26.6 | 42.0 | 20.1 | 39.2 |
> | mPLUG-Owl3 | 28.3 | 35.9 | 32.1 | 35.1 |
> | InternVL-2.5 | 29.1 | 42.6 | 36.1 | 40.7 |
> | LLaVA-1.5 | 26.2 | 35.7 | 28.4 | 35.4 |
> | **NAUTILUS**(LLaVA-1.5) | 27.7(+1.5) | 37.6(+1.9) | 33.2(+4.8) | 36.4(+1.0) |
> | Qwen2.5-VL | 30.6 | 43.6 | 36.6 | 42.5 |
> | **NAUTILUS**(Qwen2.5-VL) | 35.8(+5.2) | 44.4(+0.8) | 40.8(+4.2) | 42.9(+0.4) |
>
> Besides, to evaluate the generalization performance of our methods, we also conduct experiments on InternVL-2.5 and Qwen2.5-VL. As shown in the following table, the VFE module brings consistent improvements on almost all tasks, demonstrating remarkable generalization capabilities.
>
> | Method | Coarse-grainedClassification | Fine-grainedClassification | Image Caption | Region Caption | Counting | Grounding | Detection | VQA |
> | --- | --- | --- | --- | --- | --- | --- | --- | --- |
> |  | acc ↑ | acc ↑ | meteor ↑ | meteor ↑ | MAE ↓ | PR\@0.5 ↑ | AP\@0.5 ↑ | meteor ↑ |
> | InternVL-2.5 | 91.3 | 85.7 | 0.206 | 0.107 | 42.1 | 37.4 | 24.6 | 0.350 |
> | **NAUTILUS**(InternVL-2.5) | 92.0(+0.7) | 87.7(+2.0) | 0.206(+0.0) | 0.108(+0.001) | 38.2(+3.9) | 38.4(+1.0) | 26.7(+2.1) | 0.353(+0.003) |
> | Qwen2.5-VL | 89.1 | 92.1 | 0.220 | 0.189 | 40.7 | 37.6 | 41.9 | 0.357 |
> | **NAUTILUS**(Qwen2.5-VL) | 93.1(+4.0) | 92.3(+0.2) | 0.221(+0.001) | 0.189(+0.0) | 35.3(+5.4) | 40.3(+2.7) | 44.8(+2.9) | 0.354(-0.003) |
>
> - **To Weakness 4**: “The study would be strengthened by an ablation that compares this method to other, more naive weighting strategies to justify its design choice.”
>
> **Reply:** Good suggestion! Following your advice, we conduct ablation studies on the weighting strategies, with results presented in the following table. In particular, we explore four designs of weighting strategies, including “wo/ weighting”, “a norm weight e”, “Learned from image feature”, and “Learned from depth feature (ours)”. Among them, the “wo/ weighting” strategy can be considered as “a norm weight 1”. It achieves similar performance with the “a norm weight e” strategy, slightly inferior to other learning-based strategies. The “Learned from image feature” strategy, which means a loss of depth information, presents comparable performance compared with our design in the image caption, region caption, and VQA tasks. However, depth information is essential for distance perceptions of complex underwater objects, benefiting underwater scene understanding intuitively. Our strategy with the fusion of depth features surpasses other strategies, indicating the effectiveness of our design choice. We will add this ablation study in the revised version.
>
> | Weighting Strategy | Coarse-grained Classification | Fine-grained Classification | Image Caption | Region Caption | Counting | Grounding | Detection | VQA |
> | --- | --- | --- | --- | --- | --- | --- | --- | --- |
> |  | acc ↑ | acc ↑ | meteor ↑ | meteor ↑ | MAE ↓ | PR\@0.5 ↑ | AP\@0.5 ↑ | meteor ↑ |
> | wo/ weighting | 88.3 | 88.5 | 0.213 | 0.181 | 49.8 | 36.8 | 37.1 | 0.341 |
> | a norm weight $e$  | 88.4 | 87.6 | 0.211 | 0.185 | 50.0 | 37.0 | 36.0 | 0.346 |
> | Learned from image feature | 90.7 | 91.8 | **0.221** | 0.188 | 35.9 | 37.5 | 40.6 | **0.364** |
> | Learned from depth feature (ours) | **93.1** | **92.3** | **0.221** | **0.189** | **35.3** | **40.3** | **44.8** | 0.354 |

---

> > ### Comment · Reviewer_R364 · 2025-08-06
> >
> > Thanks for the author's reply. For w1 which addressed my concern, I strongly recommend to supplete it into the final version which strongly strenthen this paper about the robustness aspect. For w2, the reviewer suggests the author to include more visual qualitative results in the next version. The reviewer will maintain the current score.

---

> > > ### Author Response · Authors · 2025-08-07
> > >
> > > Dear Reviewer R364,
> > >
> > > We appreciate your constructive comments. We will integrate the full discussions as suggested, which we agree will significantly strengthen the manuscript. Thank you for maintaining your positive evaluation of our work.
> > >
> > > Best regards,
> > >
> > > Paper 3623 Authors

---

### Official Review · Reviewer_5R6T · 2025-07-04

**Clarity:** 3
**Significance:** 3
**Originality:** 2
**Rating:** 4
**Confidence:** 4

**Summary:**

This paper introduced an underwater large vision-language model, namely, NAUTILUS, alongside a new large-scale dataset, i.e., NautData, comprising 1.45 million image-text pairs covering eight underwater scene understanding tasks. It tackled the long-standing challenge of underwater image degradation by incorporating a physically grounded Vision Feature Enhancement (VFE) module based on underwater imaging principles. The authors integrated the VFE into two LMM baselines (i.e., LLaVA-1.5 and Qwen2.5-VL). The experimental results demonstrated consistent improvements over strong baselines on multiple datasets.

**Questions:**

1. Can the VFE module be plugged into other LMM architectures? This would clarify its universality.
2. What is the runtime overhead introduced by the VFE module?
3. Will the trained checkpoints and data annotation tools be released on the acceptance of the paper? This is of great significance for the community’s reproduction and further development.
4. How does NAUTILUS handle domain shifts between shallow and deep-sea environments? Have the authors tested the model across varying light/color conditions?
Positive responses or additional evidence addressing the first three questions would increase my confidence in the model’s robustness and practical applicability, and potentially raise my score.

**Ethical Concerns:**

["NO or VERY MINOR ethics concerns only"]

**Final Justification:**

I think that the authors have responded to most of my comments. Thanks. I will maintain my scores.

**Limitations:**

Yes

**Quality:**

2

**Strengths And Weaknesses:**

Strengths:
1. The VFE module was well-motivated by the physics-based imaging theory and effectively improved robustness under underwater image degradation.
2. The paper was clearly written and well-organized. Figures effectively supported the technical content.
3. This work made a high-impact contribution to a critical domain—underwater vision-language modeling. Both NAUTILUS and NautData could form a strong foundation for future research and applications in marine robotics, ocean monitoring, and scientific exploration, given that the authors would make them publicly available.

Weaknesses:
1. Although zero-shot evaluations were provided on MarineInst20M, more cross-domain generalization tests would strengthen the claims.
2. The VFE module involved depth extraction and attention mechanisms. A runtime/memory cost analysis was missing. This is important for practical deployment on underwater devices.

---

> ### Author Rebuttal · Authors · 2025-07-31
>
> Thanks for providing feedback and taking the time to review our work! We promise that the training/inference codes, checkpoints, and the entire NautData, alongside the data annotation tools, will be released.
>
> **Weaknesses:**
>
> - **To Weakness 1 & Question 4**: “Although zero-shot evaluations were provided on MarineInst20M, more cross-domain generalization tests would strengthen the claims.” / “How does NAUTILUS handle domain shifts between shallow and deep-sea environments? Have the authors tested the model across varying light/color conditions?”
>
> **Reply:** Thanks! Following your advice, we additionally provide zero-shot evaluations on UIIS10K [1] and TrashCan [2]  datasets, which are **never involved in our training data**, to test the cross-domain capabilities of our models. Specifically, both of them only provide bounding box annotations. We randomly sample 200 images from each of these two datasets and employ the Gemini 2.5 Flash model to generate descriptions of each instance. Subsequently, our team manually verifies each description to ensure its accuracy. We train models on our NautData and directly evaluate their zero-shot grounding performance on the sampled data. The results demonstrate that the NAUTILUS surpasses other SOTA baselines by a notable margin, presenting the effectiveness of our methods.
>
> | Method | UIIS10K [1] | TrashCan [2] |
> | --- | --- | --- |
> | MiniGPTv2 | 43.5 | 19.5 |
> | mPLUG-Owl3 | 53.5 | 31.5 |
> | InternVL-2.5 | 56.0 | 41.7 |
> | LLaVA-1.5 | 51.5 | 24.5 |
> | **NAUTILUS**(LLaVA-1.5) | 53.0(+1.5) | 29.0(+4.5) |
> | Qwen2.5-VL | 62.5 | 47.5 |
> | **NAUTILUS**(Qwen2.5-VL) | 65.5(+3.0) | 49.0(+1.5) |
>
> Shallow and deep-sea environments can often be distinguished by low-light and normal-light conditions. We also test our models across varying light and color conditions to evaluate the ability of our NAUTILUS to handle domain shifts. Specifically, we employ Gemini 2.5 Flash to filter for low-light,  normal-light, green-color, and blue-color images from the NautData test set. Then, we separately evaluate performance on each of these subsets. It is worth noting that our method presents +5.2 and +4.2 PR\@0.5 improvements while suffering significant image degradations in low-light and green-color scenarios, respectively, demonstrating the remarkable robustness and practical applicability of our methods. We will add these experiments to our revised version.
>
> | **Method** | **Low-light** | **Normal-light** | **Green-color** | **Blue-color** |
> | --- | --- | --- | --- | --- |
> | MiniGPTv2 | 26.6 | 42.0 | 20.1 | 39.2 |
> | mPLUG-Owl3 | 28.3 | 35.9 | 32.1 | 35.1 |
> | InternVL-2.5 | 29.1 | 42.6 | 36.1 | 40.7 |
> | LLaVA-1.5 | 26.2 | 35.7 | 28.4 | 35.4 |
> | **NAUTILUS**(LLaVA-1.5) | 27.7(+1.5) | 37.6(+1.9) | 33.2(+4.8) | 36.4(+1.0) |
> | Qwen2.5-VL | 30.6 | 43.6 | 36.6 | 42.5 |
> | **NAUTILUS**(Qwen2.5-VL) | 35.8(+5.2) | 44.4(+0.8) | 40.8(+4.2) | 42.9(+0.4) |
>
> [1] UWSAM: Segment Anything Model Guided Underwater Instance Segmentation and A Large-scale Benchmark Dataset. arXiv 25.
>
> [2] TrashCan: A Semantically-Segmented Dataset towards Visual Detection of Marine Debris. arXiv 20.
>
> - **To Weakness 2 & Question 2**: “The VFE module involved depth extraction and attention mechanisms. A runtime/memory cost analysis was missing. This is important for practical deployment on underwater devices.” / “What is the runtime overhead introduced by the VFE module?”
>
>
>
> **Reply:**  Thanks. We analyze the runtime and memory cost of our NAUTILUS on a single NVIDIA RTX 4090 GPU. Specifically, we fix the resolution of an input image as 336$\times$336 and observe that the runtime of the depth encoder and VFE module are 17.48 ms and 22.71 ms, respectively, each consuming 1.07 M and 1.31 M GPU Memory. We will add the runtime and memory cost analysis in the revised version.
>
> | Component | Runtime (ms) | GPU Memory (M) |
> | --- | --- | --- |
> | Depth encoder | 17.48 | 1.07 |
> | VFE module | 22.71 | 1.31 |
>
>
> - **To Question 1**: “Can the VFE module be plugged into other LMM architectures? This would clarify its universality.”
>
>
>
> **Reply:** Thanks! We have plugged the VFE module into Qwen2.5-VL and LLaVA-1.5, demonstrating effectiveness based on these two LMM architectures. To evaluate its universality more comprehensively, we further plug the VFE module into InternVL-2.5 [1]. In particular, we conduct training on a third of the NautData dataset and evaluations on eight tasks to enable an extensive comparison. As shown in the following table, the VFE module brings 2.0 acc, 3.9 MAE, and 2.1 AP\@0.5 improvements on the fine-grained classification, counting, and detection tasks, respectively, presenting a remarkable universality of this method. We will add experiments and discussions in the revised version.
>
> | Method | Coarse-grained Classification | Fine-grained Classification | Image Caption | Region Caption | Counting | Grounding | Detection | VQA |
> | --- | --- | --- | --- | --- | --- | --- | --- | --- |
> |  | acc ↑ | acc ↑ | meteor ↑ | meteor ↑ | MAE ↓ | PR\@0.5 ↑ | AP\@0.5 ↑ | meteor ↑ |
> | InternVL-2.5 | 91.3 | 85.7 | 0.206 | 0.107 | 42.1 | 37.4 | 24.6 | 0.350 |
> | **NAUTILUS**(InternVL-2.5) | 92.0(+0.7) | 87.7(+2.0) | 0.206(+0.0) | 0.108(+0.001) | 38.2(+3.9) | 38.4(+1.0) | 26.7(+2.1) | 0.353(+0.003) |
>
> [1] Expanding Performance Boundaries of Open-Source Multimodal Models with Model, Data, and Test-Time Scaling. arXiv 24.
>
> - **To Question 3**: “Will the trained checkpoints and data annotation tools be released on the acceptance of the paper? This is of great significance for the community’s reproduction and further development.”
>
> **Reply:** Thanks! We promise to release the training/inference codes, trained checkpoints, and data annotation tools, facilitating further development of this community.

---

> > ### Comment · Reviewer_5R6T · 2025-08-07
> >
> > Thanks for addressing most of my comments. I will maintain my scores.

---

> > > ### Author Response · Authors · 2025-08-07
> > >
> > > Dear Reviewer 5R6T,
> > >
> > > We're glad we addressed your concerns and appreciate your positive evaluation of our work. We would like to thank you again for your valuable suggestions, which helped improve our paper.
> > >
> > > Best regards,
> > >
> > > Paper 3623 Authors

---

### Decision · Program_Chairs · 2025-09-17

**Decision:**

Accept (poster)

**Comment:**

This paper presents NAUTILUS, a vision-language model specifically designed for underwater scene understanding. Also, they propose a large-scale dataset called NautData, comprising 1.45M image-text pairs across eight tasks. The core contribution is the Vision Feature Enhancement module, which improves robustness under low-visibility conditions and can be seamlessly integrated into mainstream multimodal models. While reviewers initially raised concerns about generalization, the motivation for feature-space enhancement, and the design of the dark-pixel prior, the authors addressed these points with extensive rebuttal experiments, including cross-dataset zero-shot evaluations, robustness tests under varying lighting and turbidity, and human-rated dataset quality assessments. Following the rebuttal, most reviewers shifted to a positive stance. Thus, the AC recommends accepting this paper.